# Plasma membrane recycling drives reservoir formation during *Toxoplasma gondii* intracellular replication

Julia von Knoerzer-Suckow[1☉], Eva-Helena Aden[1☉], Romuald Haase[2], Andreas Klingl[3], Ignasi Forné[4], Simon Gras [1]*

1 Chair of Experimental Parasitology, Ludwig-Maximilians-University Munich, Planegg-Martinsried, Germany, 2 Department of Microbiology and Molecular Medicine, Faculty of Medicine, University of Geneva, Geneva, Switzerland, 3 Plant Development, Ludwig-Maximilians-University Munich, Planegg-Martinsried, Germany, 4 Biomedical Center (BMC), Protein Analytics Core Facility, Faculty of Medicine, LMU Munich, Munich, Germany

☉ These authors contributed equally to this work.

* simon.gras@lmu.de

## Abstract

During intracellular development, apicomplexan parasites reside within a parasitophorous vacuole largely derived from the host plasma membrane (PM) and rendered nonfusogenic with the host endolysosomal system. Yet, the parasite is capable of protein uptake from the host cell via endocytosis, which occurs via a conserved structure, the micropore. Recently the composition of the micropore was characterized and its stability was shown to depend on the presence of the kelch-domain protein K13 which is also central to malarial drug-resistance to artemisinin. Interestingly, depletion of K13 also resulted in an impressive accumulation of PM attached to or between individual parasites, suggesting that the micropore plays a critical role in PM homoeostasis. Here, we characterized the dynamics and recycling of the PM in *Toxoplasma gondii*. In intracellular parasites, the PM is shared between individual parasites and undergoes a cycle of endocytosis and exocytosis during replication, similar to what has been previously demonstrated for extracellular parasites. This cycle appears to depend on Rab5b and MyoF. Interestingly, in contrast to *Plasmodium falciparum*, Rab5b is dispensable for the lytic cycle of *T. gondii*. During replication, parasites establish an extracellular plasma membrane reservoir (PMR) prior to daughter cell formation. The PMR is a dynamic membranous structure that varies in size and position throughout replication and disappears after daughter cell budding. Perturbation of the endo-exocytic balance disrupts PMR formation, leading to increased number and size of PMRs and, ultimately, to a complete loss of membrane organization directly linking endocytosis to the regulation of PMR formation.

**Data availability statement:** All relevant data are within the paper and its Supporting information file S1 and S2 Data.

**Funding:** This work was funded by the Deutsche Forschungsgemeinschaft (DFG, German Research Foundation, https://www.dfg.de/en) with a research grant to SG: grant number GR 5696/2-1 and an equipment grant INST 86/1831-1 to Principal Investigator Markus Meissner. It was also funded by the Swiss National Science Foundation (https://www.snf.ch/en) with a grant to Principal Investigator Dominique Soldati-Favre: grant number: 310030_215445. The funders had no role in study design, data collection and analysis, decision to publish, or preparation of the manuscript.

**Competing interests:** The authors have declared that no competing interests exist.

**Abbreviations:** APS, ammonium persulfate; ATC, anhydrotetracycline; BSA, bovine serum albumin; Cb, Chromobody; DG, dense granules; DN, dominant-negative; ELCs, endosome-like compartments; Endo-SAG1, endocytosed SAG1; ESCRT, Endosomal Sorting Complex Required for Transport; FDR, false discovery rate; FI, fluorescence intensity; FOV, field of view; FRAP, fluorescence recovery after photobleaching; H.O.S.T., Host Organelle–Sequestering Tubulostructures; HCCU, host cell cytosol uptake; HFF, human foreskin fibroblast; IMC, inner membrane complex; mAID, mini-auxin-inducible degron; MNP, membrane nonpermeable; MP, Membrane permeable; n-SAG1, de novo SAG1; PFA, paraformaldehyde; PM, plasma membrane; PMR, plasma membrane reservoir; TEMED, tetramethylethylenediamine; TGN, trans-Golgi network.

## Introduction

The Apicomplexa phylum comprises parasites of veterinary and medical importance, such as *Plasmodium falciparum*, *Cryptosporidium parvum*, and *Toxoplasma gondii* [1–3]. These parasites are obligate intracellular parasites that establish a parasitophorous vacuole [4,5], during the active invasion process [6]. Furthermore, parasites secrete numerous factors to modify the host cell, including the hijacking of trafficking factors, such as the Endosomal Sorting Complex Required for Transport (ESCRT) or the Host Organelle–Sequestering Tubulostructures (H.O.S.T.) [7,8].

The uptake of host material is critical for the parasite's development as it provides them with nutrients [9–12]. However, in the case of *T.gondii*, while the uptake of host nutrients has been well demonstrated, the role of endocytosis of host proteins, although demonstrated in several recent studies remains unclear [7]. Only recently the role of the micropore in this process was demonstrated [13,14]. This structure is integrated into the inner membrane complex (IMC) of the parasite, providing an interface between the plasma membrane (PM) and the cytosol. It is composed of a ring containing the proteins K13, Eps15L, MCA3, CGAR, and ISAP1. Inside this ring, a funnel composed of KAE, AP-2α, AP-2μ, AP-1/2β, AP-2σ, AGFG, and UBP1 allows the passing and endocytosis of the PM [13]. Interestingly, while the studies of Koreny and colleagues [13] and Wan and colleagues [14] agree concerning the composition of the micropore, when it comes to the uptake of host cell proteins, they reach different results and conclusions. While Koreny and colleagues [13] were unable to see a significant difference in the uptake of host cell proteins upon disruption of the micropore, Wan and colleagues [14] suggest that the micropore is required for this uptake. However, in both cases, uptake and recycling of PM through the micropore has been demonstrated, and Koreny and colleagues [13] speculated that this structure, besides its role in nutrient uptake, might also be required for PM homeostasis during parasite replication.

To address this hypothesis, we set out to analyze the dynamics and the recycling of the PM during parasite development and investigated the trafficking pathway of the endocytosed vesicles. Furthermore, parasite behavior upon disruption of K13 and previously described trafficking factors, such as the unconventional myosin MyoF was analyzed in detail for their role in endocytosis.

To measure membrane dynamics, we optimized a dual color labeling assay, based on tagging the major surface protein, the GPI-anchored SAG1 with the HALO-tag [13]. This novel strategy allows the differentiation of surface and internally localized SAG1 as well as de novo-synthesized SAG1 during replication. Using this strategy, we demonstrate that [1] an endo-exocytic cycle is active during replication supporting the inheritance of maternal PM [2] parasites start to form an extracellular PM reservoir at the onset of replication that is reabsorbed after the hatching of daughter parasites from the mother and [3] depletion of K13 leads to a drastic reduction of the endocytosed vesicles and accumulation of the PM reservoirs, leading to morphological abnormal parasites.

## Results

### Diffusion of membrane content between parasites of the same vacuole

The PM is composed of lipids and proteins and serves as a boundary between the intracellular and extracellular environments [15]. To visualize PM dynamics during replication, we initially tested several lipid dyes. However, all the tested dyes demonstrated high background and nonspecific labeling, as shown for Nile red (S1 Fig). Therefore, we decided to use the major surface antigen SAG1, a GPI-anchored protein as a proxy for PM dynamics.

Like most apicomplexans, *Toxoplasma* tachyzoites appear very rigid in shape, as no deformation of the cell can be observed when the parasite is gliding on substrates [16,17], in contrast to most eukaryotic cells [18,19]. This appears to be a consequence of the PM being directly connected to the IMC, a unique organelle underlying the parasite's PM. The IMC consists of flattened rectangular membrane sacs that are connected to the cytoskeleton and at the same time to the PM via the actomyosin motor, in particular via the gliding associated protein GAP45 [20,21]. While this connection stabilizes the overall shape of the parasite, the PM appears to be highly fluid, as demonstrated by beads, which are translocated at the surface even in the absence of motility [16]. To test if SAG1 is a good proxy for the analysis of PM behavior, we performed FRAP (fluorescence recovery after photobleaching) experiments on parasites expressing SAG1-Halo [22,23] (Fig 1). Parasites were labeled with Alexa-488, a membrane nonpermeable (MNP) dye. Two types of FRAP experiments were designed on freshly invaded parasites: one, to address the fluidity of the membrane at a local scale, and another to address the global fluidity of the PM (Fig 1A and 1D).

First, we performed FRAP on a small area of the PM (Fig 1B). The bleaching of the PM led to a drop of 74% of the relative signal intensity. This reduction in fluorescence intensity (FI) was followed by a quick recovery in ~20 s. The recovered signal within the bleached area reached the same intensity as nonbleached areas of the PM (Fig 1C). This rapid recovery demonstrates that the PM is highly fluid. Furthermore, it validates SAG1-Halo as a good proxy for PM-dynamics, since it appears to freely diffuse within the PM.

To address the fluidity at the scale of the whole PM, the FRAP area was increased, and half of the PM was bleached (Fig 1E). As observed previously, the FI of the signal at the bleaching point dropped from 100% to 22% after FRAP. A FI recovery was observed, increasing from 22% to 40%. It also required a longer time to recover compared to the previous experiment (~40 s) (Fig 1F, blue line). In parallel, the PM FI at the opposite side of the FRAP area was also impacted with a drop of the FI of 28% when the bleaching was performed. Interestingly, it was followed by a gradual decrease of the FI down to ~50% of the initial FI. These results suggest that SAG1 diffuses over the whole PM. Taken together the data show that the PM of *T.gondii* has a high fluidity and that SAG1 can be used as a marker to measure PM behavior of the parasite.

During replication, it has been demonstrated that daughter cells remain connected via an intravacuolar network, which is essential for the exchange of material and signals between individual parasites [24]. The central point of this connection is referred to as the residual body [25]. This residual body, along with the connection between parasites, is supported by F-actin filaments, which serve multiple functions, ranging from parasite synchronization to organelle trafficking. These F-actin filaments were visualized using camelid nanobodies called Chromobody (Cb), expressed inside the parasite [26]. FRAP experiments on vacuoles containing multiple parasites (expressing a cytosolic GFP), showed a fast recovery after bleaching and therefore efficient exchange of material from one parasite within the PV to another [26,27]. This sharing of cytoplasmic content is lost in mutant parasites where the intravacuolar network is absent, for example in a conditional null mutant for *act1* which cannot form the F-actin filaments [28].

As previously demonstrated, parasites within a single PV are connected at the posterior end, with a continuous PM and a shared cytoplasm (Fig 1G). The PM also covers the newly formed residual body where F-actin filaments are located (Figs 1G and S2). This coverage of the residual body is maintained throughout replication, as demonstrated by the colocalization of SAG1 and F-actin filaments. The F-actin filaments were imaged using an actin-chromobody fused to a SNAP tag (Cb-SNAP). The SNAP tag technology, similar to the Halo tag, allows visualization in different colors using specific

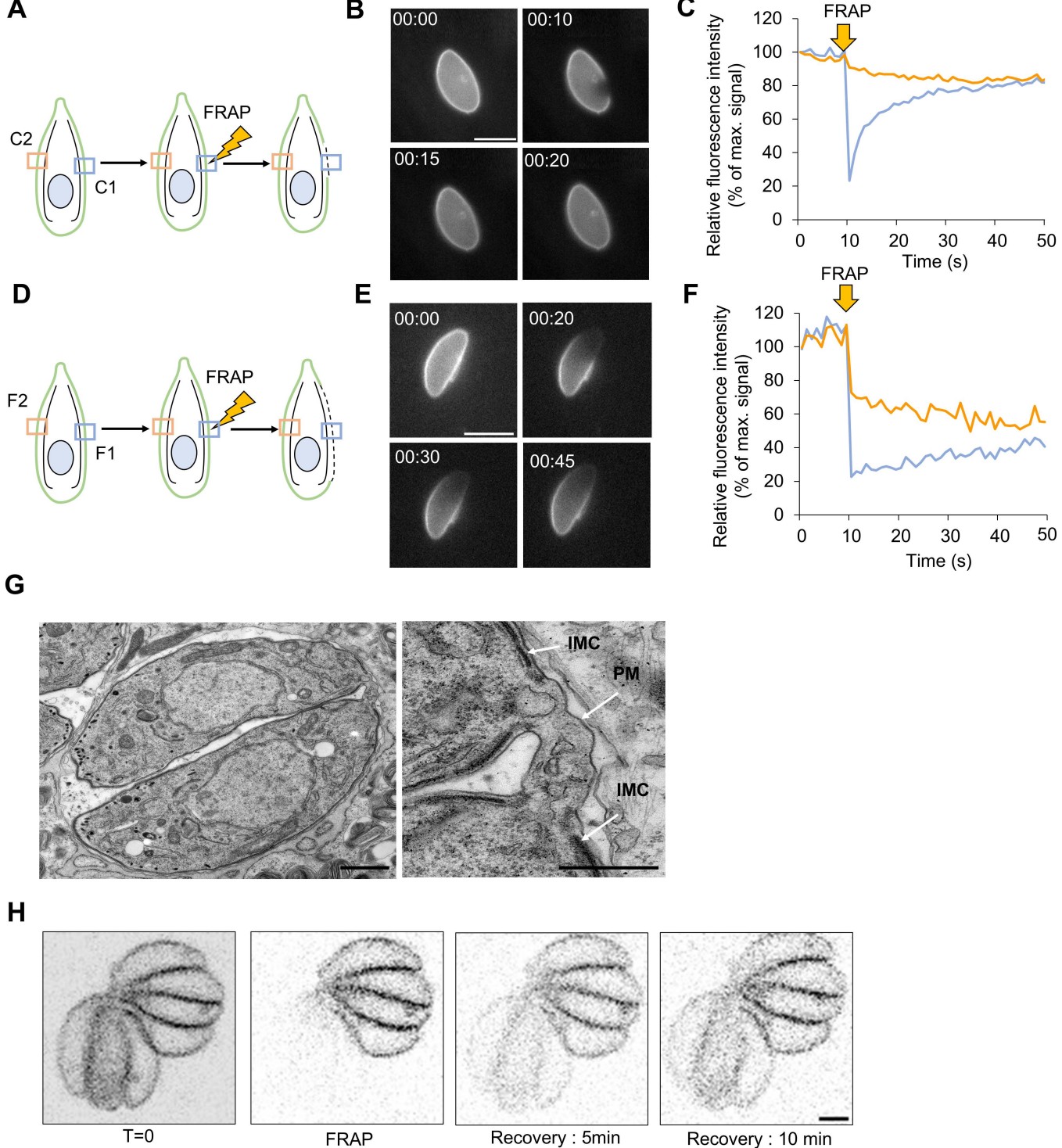

**Fig 1. *Toxoplasma gondii* forms a syncytium with dynamics exchange at the plasma membrane.** For the FRAP experiments, SAG1-Halo parasites were labeled with the membrane nonpermeable dye Alexa-488. **A)** Schematics of the small area FRAP experiments performed on intracellular single parasites with the indicated recorded region of their corresponding curves. C1 in blue and C2 in yellow. **B)** Time series images from a parasite with a small area FRAP. Time in seconds. Scale bar 5 μm. **C)** Relative fluorescence intensity curves of the small area FRAP. Values are expressed as a percentage of the fluorescence recorded before FRAP and are normalized to control parasites. **D)** Schematics of the large area FRAP experiments

performed on intracellular single parasites with the indicated recorded region of their corresponding curves. F1 in blue and F2 in yellow. **E)** Time series images from a parasite with a large area FRAP. Time in seconds **F)** Relative fluorescence intensity curves of the large area FRAP. Values are expressed as a percentage of the fluorescence recorded before FRAP and are normalized to control parasites. **G)** TEM of a stage 2 vacuole. The connection between the two parasites is highlighted on the right. PM: Plasma membrane, IMC: inner membrane complex. Scale bars 1 µm and 500 nm. **H)** FRAP experiment on large stage vacuole. Half of the parasites of the vacuoles were bleached, and recovery was recorded for 10 min. Scale bar 1 µm. The data underlying this figure can be found in S1 Data.

dyes. We observed in 93 ± 6% of the vacuoles, even at later replication stages (S2B and S2C Fig). This maintained connection should allow the transfer of proteins between parasites within the same vacuole.

To assess the potential exchange at the PM between parasites of the same vacuole, FRAP experiments were performed on large vacuoles, where half of the parasites were bleached (Fig 1H). We observed recovery of the SAG1 signal at the surface within 5–10 min. These results illustrate that the membrane is fluid between individual parasites of the PV. The longer recovery time, when compared to the experiments on individual parasites can be explained by the intravacuolar network, potentially representing a bottleneck for diffusion. Nevertheless, this experiment demonstrates that parasites within a PV share the same PM content.

### Inheritance of the maternal PM material during replication

Toxoplasma replicates in a process, called endodyogeny, where two daughter parasites grow within the mother and finally hatch out, raising the question of the inheritance of the maternal PM. To precisely characterize the PM behavior, we defined five different PM fractions using SAG1-Halo localization and production time as reference: (1) the PM at the surface of the parasite, represented by the SAG1-Halo facing the extracellular environment (PM-SAG1); (2) the PM not yet secreted, still in form of internal vesicles inside the parasite cytoplasm (Int-SAG1); (3) the maternal PM, which include the fractions 1 and 2 of the mother cell PM before replication (M-SAG1); (4) the de novo PM which is generated during replication (n-SAG1); and (5) the endocytosed PM in form of vesicles (endocytosed SAG1, Endo-SAG1).

Using a variety of different staining schemes, relying on the versatility of the Halo tag, we set to characterize the behavior of these different PM fractions during replication (Fig 2A).

To follow PM-SAG1 during replication, freshly egressed parasites were labeled for 1 h with a membrane nonpermeable (MNP) dye prior invasion and replication (Figs 2A, Top and S3A). Twenty-four hours after the maximal FI of PM-SAG1 was then measured for vacuoles at different replication stages, 1–8. (Fig 2B and 2E). After each replication, the FI was reduced by ~2-fold, passing from 100% at stage one to 53 ± 3% at stage 2, then 30 ± 6% at stage 4, and finally 18 ± 4% at stage 8. Considering that the overall PM surface doubles after each round of replication, these results illustrated that PM-SAG1 is efficiently recycled from the mother to daughter parasites. From this result several hypotheses are possible: (1) the maternal PM stays intact at the surface of the mother cell and is taken over by the hatching daughters at the end of endodyogeny; (2) maternal PM is inherited by the daughters in a process requiring a cycle of endocytosis and recycling back to the surface, or (3) a combination of 1 and 2. Since, we also identified uptake of PM-SAG1 (Fig 2B), we suggest that inheritance of maternal PM follows most likely option 3.

During the growth and replication of parasites de novo generation of PM is required. To track de novo synthesis, freshly egressed parasites were labeled for 1 h with a membrane-permeable dye before invasion and replication, labeling and saturating all M-SAG1 (Figs 2A, Middle and S3B). After 24 h of replication, n-SAG1 was labeled using a new membrane-permeable dye of a different color. FI of both M-SAG1 and n-SAG1 was measured for parasites at different stages of replication as described above (Fig 2C). M-SAG1 FI followed, in a less linear manner, the ~2-fold decrease observed for PM-SAG1, passing from 100% to 73 ± 18% from stage 1 to 2, then 35 ± 6% at stage 4 and 24 ± 2% at stage 8 (Fig 2F, magenta). In opposition, n-SAG1 FI increased with each replication step passing from 30 ± 11% to 63 ± 5% at stage 2 and reaching a plateau of 91 ± 14% at stage 4 and 88 ± 16% at stage 8 (Fig 2F, green). Since the total amount of SAG1 at the

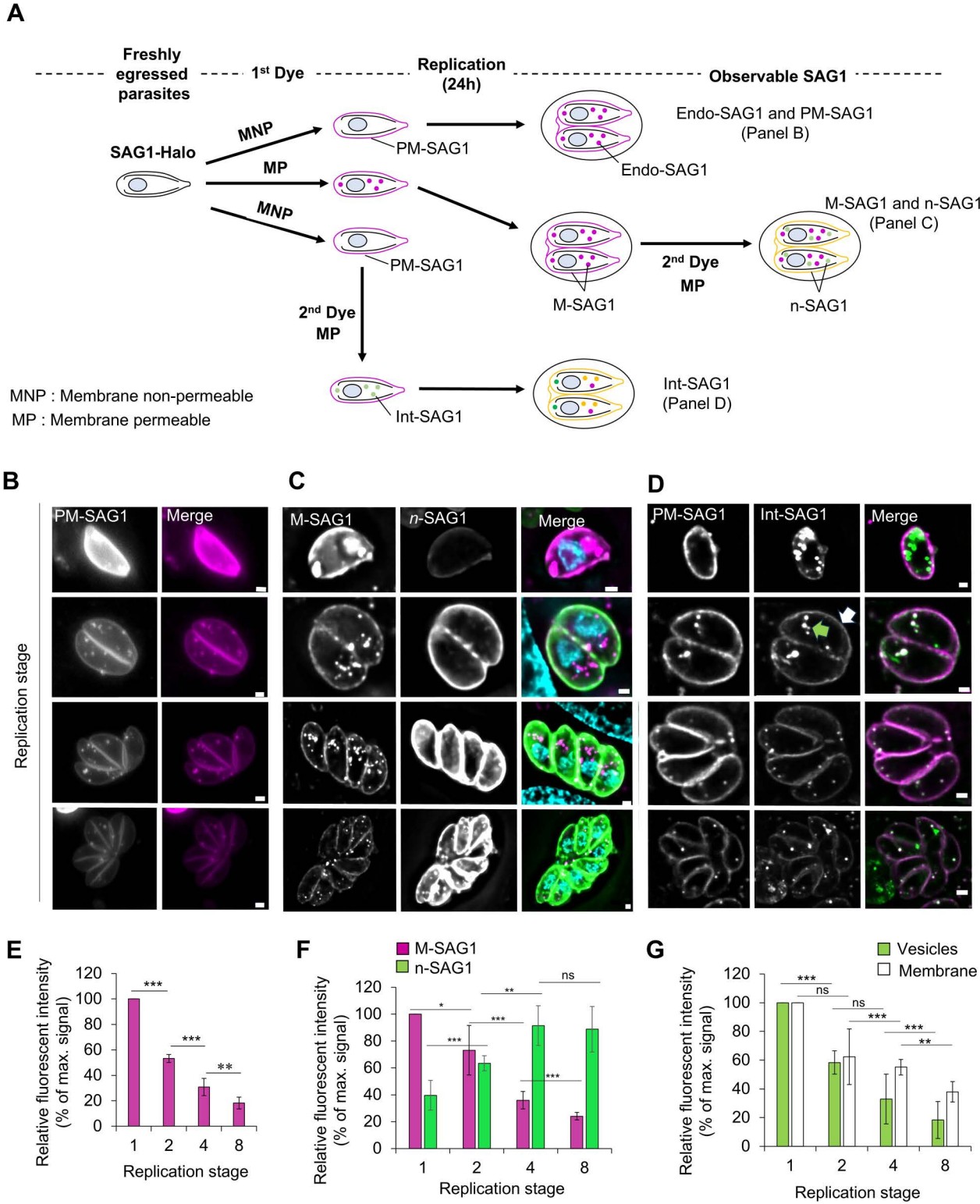

**Fig 2. The cycle of recycling and de novo generation of the plasma membrane during endodyogeny. A)** Schematics of the 3 labeling experiments, performed to observe the different SAG1-Halo populations. Top: extracellular labeling using a membrane nonpermeable dye to track the maternal plasma membrane SAG1: PM-SAG1. Middle: extracellular labeling using a membrane-permeable dye to track maternal SAG1: M-SAG1, the association

of both PM and Int-SAG1. After transfer to host cells and 24 h of replication, another membrane-permeable dye is used to track the newly generated SAG1: n-SAG1. Bottom: extracellular labeling using a membrane nonpermeable dye followed by a membrane permeable to track the maternal internal SAG1: Int-SAG1. Each dye is incubated for 1 h, then washed thoughtfully before transferring the parasite on host cells. **B)** Representative pictures of parasites at different stages of replication (1–8) with PM-SAG1 labeling. **C)** Representative pictures of parasites at different stages of replication (1–8) with M-SAG1 and n-SAG1 labeling. **D)** Representative pictures of the parasites at different stages of replication (1–8) with PM-SAG1 and Int-SAG1 labeling. **E)** Quantification of the fluorescence intensity of the plasma membrane PM-SAG1 during replication. **F)** Quantification of the fluorescence intensity of the maternal M-SAG1 (magenta) and newly generated n-SAG1 (green) during replication. **G)** Quantification of the fluorescence intensity of the Int-SAG1 in the vesicles (green) and at the plasma membrane (white). Three biological replicates were used for all analyses; all $P$ values are $0 \leq P \leq 0.001$, ***, error bars are standard deviations, and the center measurement of the graph bars is the mean. A one-tailed unpaired Student $t$ test was used for all comparisons with no adjustments. All scale bars = 1 μm. The data underlying this figure can be found in S1 Data.

PM should remain relatively constant, reaching this FI plateau suggests that the parasites require two rounds of replication until the majority of the PM consists of n-Sag1. Interestingly, insertion of de novo material in the PM was also observed in stage 1 parasites before replication (Fig 2C and 2F).

Since parasites appeared to have both, surface and internal SAG1 prior to the onset of replication, we were interested to determine if Int-SAG1 is delivered to the surface during replication, as previously demonstrated for extracellular parasites [13] or potentially degraded. Therefore, we decided to follow the fate of Int-SAG1 in detail. To specifically follow Int-SAG1, extracellular parasites were first labeled for 1 h with a MNP dye, leading to saturation of PM-SAG1. Parasites were washed and incubated for 1 h with a membrane-permeable dye, labeling exclusively Int-SAG1 (Figs 2A, bottom and S3C). After wash, and 24 h of replication on host cellInt-SAG1 signal was observed in intracellular vesicles and but also in the PM (Fig 2D). The FI Int-SAG1 in the vesicles was decreased by ~2-fold at each replication, passing from 100% to 58±19% between stages 1 and 2, then 32±5% at stage 4 and 18±7% at stage 8 (Fig 2G, green). This decrease in FI inside the Int-SAG1 vesicles suggests that less material is present within the vesicles. Two hypotheses could explain this reduction: (a) the material inside the Int-SAG1 vesicles is degraded, or (b) the material inside the Int-SAG1 vesicles is secreted.

To test these hypotheses, we measured the FI of Int-SAG1 located at the PM (Fig 2G, white). Similar to PM-SAG1 and M-SAG1, the intensity decreased after each replication cycle (Fig 2E). However, when comparing PM-SAG1 and Int-SAG1 signals at the PM, we observed that Int-SAG1 FI decreased more slowly than PM-SAG1 (S3D Fig). This slower decrease indicates insertion of labeled Int-SAG1 into the PM. Since the dye used to label Int-SAG1 was removed and excess dye washed off prior to transferring parasites to host cells, the observed signal must originate from intracellular vesicular stores. This supports hypothesis (b)—that Int-SAG1-containing vesicles are secreted to the PM during replication.

Taken together, these results depict a very dynamic system where both endocytosis, secretion, and inheritance of the PM occur during replication.

### Characterization of the SAG1-Halo vesicles trafficking pathway

Next, we aim to characterize the trafficking pathways used by the different populations described above. The total SAG1 pool consists of three different subpopulations: (1) SAG1 at the PM, (2) SAG1 recycled from the PM, and (3) de novo synthesized PM. The Int-SAG1 population, observed in freshly egressed parasites, likely represent a mix of Endo-SAG1 and de novo-SAG1. For the recycling of SAG1, it would therefore be expected that subpopulations (1) and (2) show good co-localization, whereas (3) should be in independent vesicles and show only co-localization, once the recycling and secretion pathway merge.

Previously, it has been demonstrated that *T. gondii* endocytic trafficking involves transit through early and late endosome-like compartments (ELCs) and potentially the *trans*-Golgi network (TGN) as in plants [29]. Furthermore, exocytic trafficking to regulated secretory organelles, micronemes, and rhoptries, also proceeds through ELCs and requires classical endocytic components [30].

To test this hypothesis, we performed a modified triple labeling strategy, allowing us to independently label these subpopulations (Fig 3A). Freshly egressed parasites were first labeled for 1 h with a MNP dye (Halo Alexa-488), resulting in

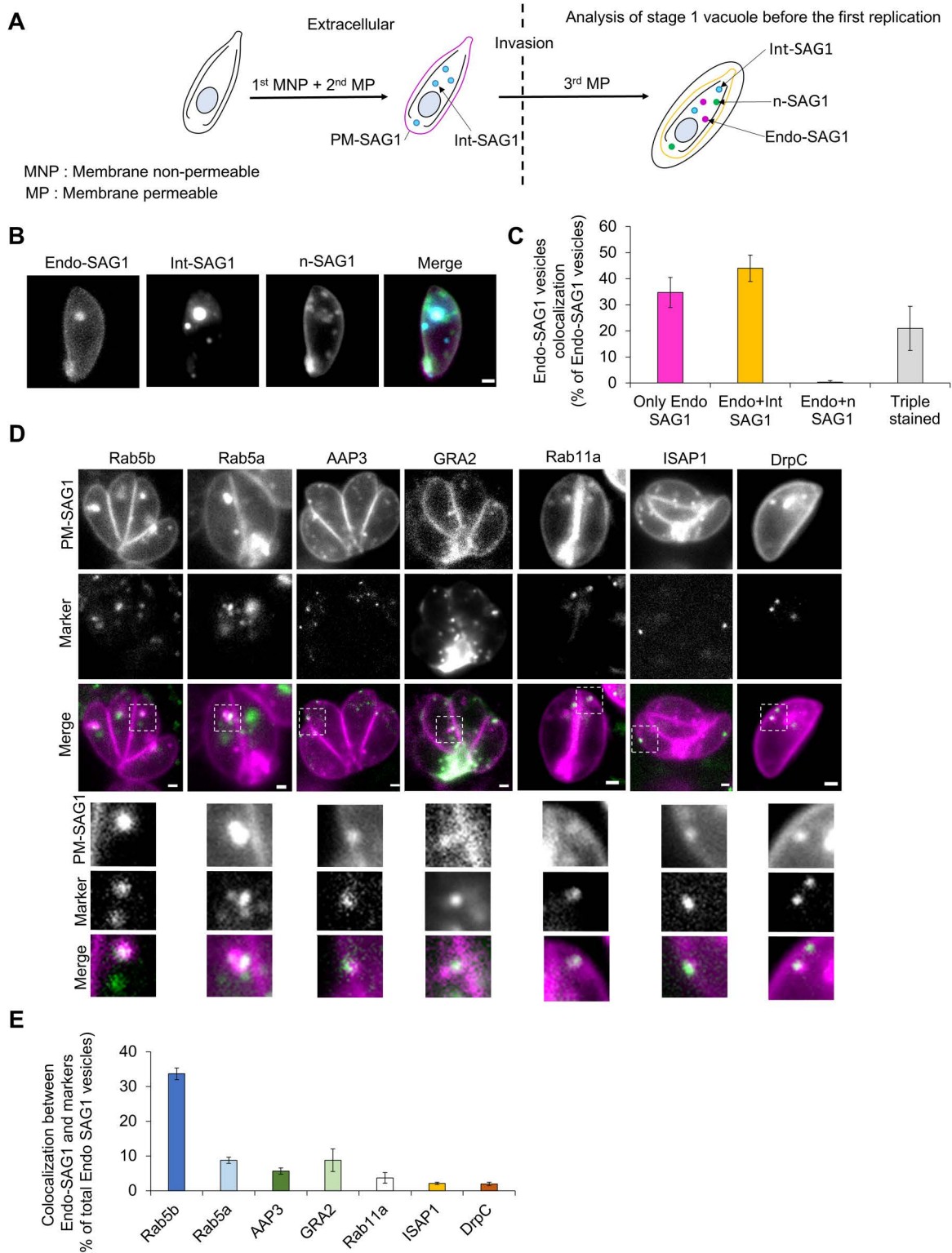

**Fig 3. Characterization of SAG1 Halo vesicles. A)** Schematics of the labeling strategy used to compare endocytosed plasma membrane SAG1: PM-SAG1 and the newly generated SAG1: n-SAG1: extracellular labeling using a membrane nonpermeable dye followed by a membrane-permeable dye to differentiate the maternal PM-SAG1 and internal SAG1: Int-SAG1. After transfer to host cells and 24 h replication, another membrane-permeable

dye is used to track n-SAG1. Only stage one vacuoles were used for the analysis to dimmish the amount of n-SAG1 entering the recycling cycle. For each replicate, 25 parasites were analyzed, the experiment was performed in triplicate, and a total of 225 Endo-SAG1, 722 Int-SAG1, and 337 n-SAG1 vesicles analyzed. **B)** Representative picture of the triple labeling. **C)** Quantification of the colocalization between the different populations of SAG1 vesicles. Magenta: Endo-SAG1 only vesicles, yellow: vesicles containing Endo-SAG1 and Int-SAG1 only, white: vesicles containing Endo-SAG1 and n-SAG1 only, gray: vesicles containing the three populations. **D)** Representative pictures of the colocalization between Endo-SAG1 vesicles (PM-SAG1) and the markers of different organelles: dense granules (GRA2), micropore (ISAP1 and DrpC), and apical annuli (AAP3) and three trafficking factors (Rab5a, b, and Rab11). The top pictures show a vacuole where colocalization was observed and the small windows focus on the colocalization with Endo-SAG1 vesicles. **E)** Quantification of the percentage of Endo-SAG1 vesicles colocalizing with the tested markers. Expressed as a percentage of the total number of vesicles. Each measurement is independent of the other. Three biological replicates were used for all analyses; all $P$ values are $0 \leq P \leq 0.001$, ***, error bars are standard deviations, and the center measurement of the graph bars is the mean. All scale bars = 1 μm. The data underlying this figure can be found in S1 Data.

labeling of PM-SAG1. After wash, labeling with a membrane-permeable dye was performed to labeled Int-SAG1 for 1 h (Halo-Jan 646). After a new round of washes, parasites were allowed to invade and replicate for 24 h. To visualize n-SAG1, intracellular parasites were labeled for 1 h with a new membrane-permeable dye (Halo-Jan 549) (Fig 3B). Single-stage parasites were selected to visualize n-SAG1 while limiting the amount which would have completed a cycle of exo-endocytosis. This method allows us to specifically visualize the Endo-SAG1 and counterstain it with the two other internal subpopulations: Int-SAG1 and n-SAG1. A total of 225 Endo-SAG1, 722 Int-SAG1, and 337 n-SAG1 vesicles were observed.

The different co-localisations of the Endo-SAG1 vesicles were analyzed: single signal (Endo only), double signal (Endo and Int-SAG1 or Endo and n-SAG1), and triple signal (Endo-Int-n-SAG1). In good agreement with our hypothesis, we saw good co-localization of Endo-SAG1 with Int-SAG1 (44 ± 5%), but not with n-SAG1 (0.3 ± 0.5%) (Fig 3C). Cases of triple labeling were observed (20 ± 8%), which indicates that some n-SAG1 was already secreted and recycled or that all the vesicles intersected during their transport back to the PM. Together, these results suggest that the endocytic and de novo pathways are independent. Similar results were obtained when the analysis was performed using the Int-SAG1 or n-SAG1 population (S4 Fig).

After confirming that Endo-SAG1 and n-SAG1 vesicles follow different pathways, we were interested to characterize the recycling pathway in more detail, by analyzing the localization of Endo-SAG1 relative to different organellar and endocytic markers. To take the dynamic nature of the observed vesicles better into account, we decided to perform live-cell imaging. Therefore, we endogenously tagged different markers of the trafficking system with YFP (Figs 3D and S4C), such as Rab5b, Rab5a, AAP3, Gra2, Rab11A, ISAP1, and DrpC. Colocalisation between these markers and Endo-SAG1 vesicles was measured (Fig 3E).

Only a few percent of vesicles colocalised with ISAP1 (2 ± 0.3%), one of the proteins of the micropore, and DrpC (2 ± 0.4%), the dynamin likely to be involved in the generation of endocytosed vesicles [13,14]. Similar results were obtained for AAP3 (5 ± 0.9%), a marker of the apical annuli. This low percentage might result from the transient nature of the interaction between the Endo-SAG1 vesicles and these structures.

A significant co-localization was observed with the small GTPase Rab5b (33 ± 1.6%), while no important co-localization was detected with Rab5a and Rab11b (8 ± 0.8% and 3 ± 1.5%, respectively). Interestingly, Rab5b has been identified as essential in *Toxoplasma* [31] and in the case of *P. falciparum* shown to be required for host cytosol uptake [32].

Interestingly, only a small overlap between SAG-vesicles and dense granules (DG) could be detected (GRA2: 8 ± 3%). This suggests that the secretory pathway of this GPI-anchored protein is not overlapping with the secretory system for DG and other secretory organelles.

Taken together these results illustrate the independence of the endocytic and de novo trafficking pathway, as well as a different trafficking than the DG. We identified Rab5b as a potential trafficking factor for recycling.

**Rab5B is dispensable for the lytic cycle of *Toxoplasma gondii* but is critical for plasma membrane recycling**

As Rab5B showed the strongest colocalization with SAG1-positive endocytic vesicles, we sought to characterize its function. Most Rab GTPases in *T. gondii* have previously been studied through the overexpression of dominant-negative (DN)

[31]. All three Rab5 paralogues—Rab5A, Rab5B, and Rab5C—were reported as essential, as their DN overexpression led to a pronounced inhibition of the lytic cycle. However, unlike Rab5A and Rab5C, which impair both rhoptry and microneme biogenesis, Rab5B-DN expression did not produce such effects, and its function has remained unclear.

To obtain a fast regulation, Rab5B was endogenously tagged with a mini-auxin-inducible degron (mAID) in a Tir1-expressing background [33]. The mAID tag was inserted at the C-terminus to preserve the N-terminal myristoylation signal [34]. Following drug selection and clone isolation, proper tag integration at the Rab5B locus was confirmed by PCR and sequencing (S5A Fig). Treatment with auxin resulted in rapid degradation of Rab5B-mAID-HA, which became undetectable by western blot within 2 h of auxin addition (Figs 4A and S5B).

To assess the functional impact of Rab5B depletion on the lytic cycle, plaque assays were performed. Surprisingly, and in contrast to the previous observations using DN constructs or in *P. falciparum* [31,32], Rab5B depletion had minimal effect on parasite fitness (Fig 4B). A modest ~20% reduction in plaque size was observed (Figs 4C and S6A), consistent with the slightly negative fitness score reported for Rab5B (−1.3) [35]. To investigate the origin of the mild growth defect, invasion and replication assays were performed. Parasites were treated with or without auxin for 2 h prior to host cell infection. Although no statistically significant difference was observed in the replication and invasion assay (Fig 4D and 4F), each biological replicate consistently showed a slight replication delay in auxin-treated parasites, with an increased proportion of vacuoles at earlier replication stages compared to the control, as illustrated by the percentage differences shown in Fig 4E.

Using a panel of antibodies, the impact of Rab5b-KD on overall parasite morphology was assessed (Fig 4G). Upon auxin-induced depletion of Rab5b, parasites displayed normal structural organization, as indicated by GAP45 staining. The apicoplast, whose disruption is known to cause a delayed death phenotype [36], also appeared intact. All tested microneme subsets (MIC2, MIC4, MIC8, and AMA1) were properly localized, with no visible defects. Similarly, rhoptries (RON2–4) exhibited a classical apical distribution. These findings suggest that the observed growth defect is not attributable to structural abnormalities in these organelles.

Having ruled out major structural defects in key organelles, the potential involvement of Rab5b in endocytic processes was next examined (Fig 5). To this end, SAG1 was endogenously tagged with HaloTag in the Rab5B-mAID strain, as previously described [13]. Parasites were labeled with a membrane-impermeable dye (Alexa Fluor 488), washed, and transferred to host cells. After 24 h of replication, endocytosis was quantified. Rab5B depletion did not affect the percentage of vacuoles containing endocytic vesicles (Fig 5A and 5B). However, a larger proportion of parasites exhibited strong accumulations of Endo-SAG1 in the residual body (Fig 5A, zoom and 5C) (Ctrl: 9 ± 2% versus KD: 28 ± 6%). Accordingly, the percentage of Endo-SAG1 vesicles located in the cytoplasm decreased (75 ± 3% versus 54 ± 2%) while their accumulation in the RB increase (25 ± 3% versus 46 ± 2%) in the absence of Rab5b (Fig 5D). Beyond altering the vesicles localization, Rab5b depletion also affected the Endo-SAG1 vesicles themselves: their number was reduced (1.7 ± 0.3% vesicles per tachyzoites for Ctrl versus 0.9 ± 0.08% for KD, Fig 5E), while their average size increase (0.18 µm$^2$ for Ctrl versus 0.30 µm$^2$ for KD, Fig 5F), suggesting potential vesicles fusion.

To investigate the impact of Endo-SAG1 vesicle alteration on PM dynamics, the FI evolution during replication was quantified as described above (Fig 2A, 2B, and 2E). While control parasites displayed a similar pattern to WT, the addition of auxin led to a faster decrease in the PM FI (stage 2: Ctrl FI 46 ± 8% versus 24 ± 2% for KD), indicating a loss of labeled SAG1 at the parasite surface (Fig 5G and 5H). Taken our previous result into consideration, we hypothesized that the absence of Rab5b impairs the recycling of endocytosed vesicles, resulting in this loss of signal. In agreement with this idea, the reduction in membrane material delivered to the PM would need to be compensated. To test this, the generation of de novo SAG1 (n-SAG1) was assessed. Double labeling was performed as previously described (Fig 2A, 2C, and 2F) to specifically visualize newly synthesized material (Fig 5I and 5J). Imaging and FI analysis showed that Rab5b depletion, while blocking Endo-SAG1 recycling, led to increased production of n-SAG1, with stage 2 parasites containing more n-SAG1 than controls (Fig 5I, zooms) and reaching the signal plateau more quickly (stage2 Ctrl FI 74 ± 1% versus 91 ± 2%, Fig 5J).

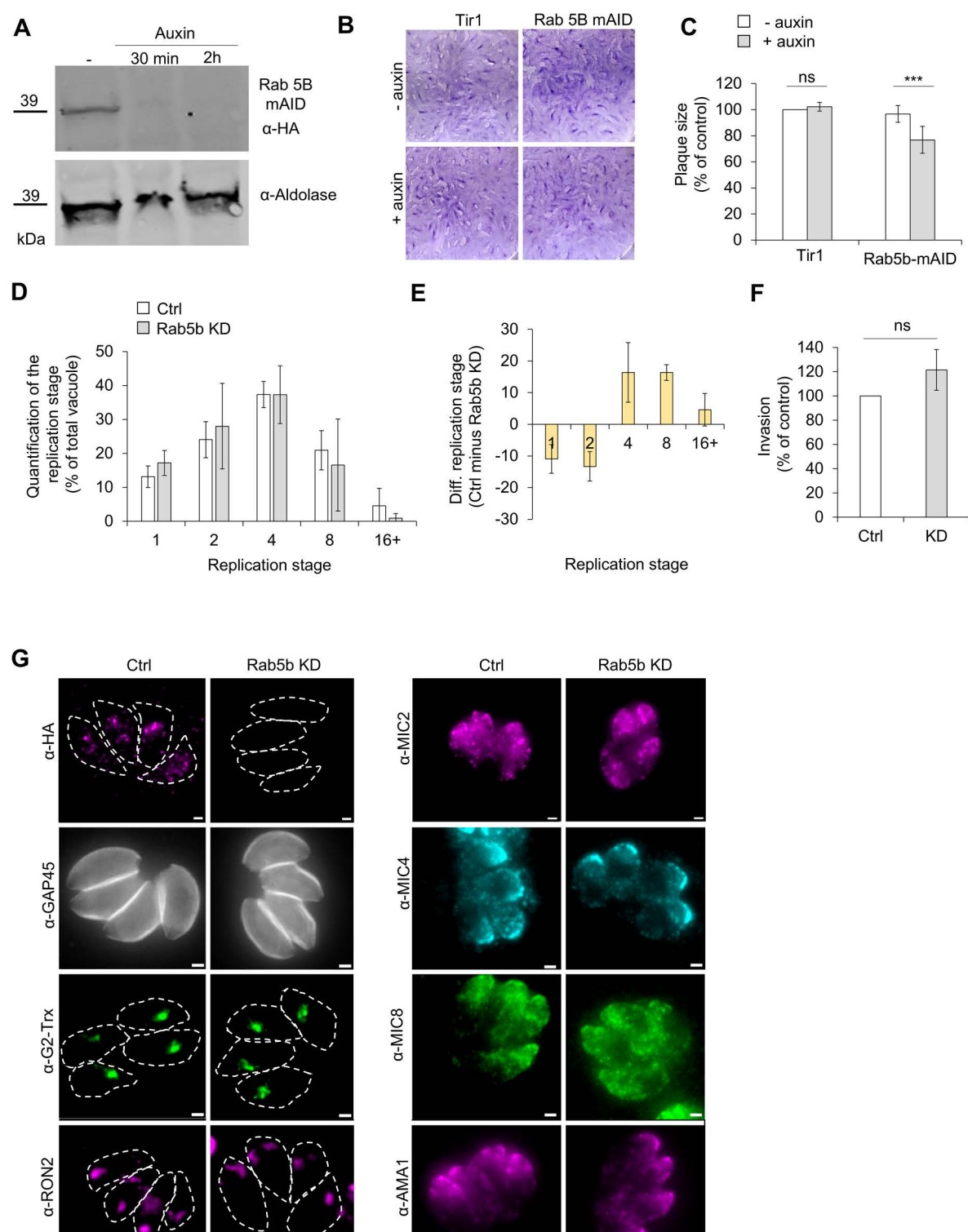

**Fig 4. Rab5b is dispensable for the lytic cycle of *Toxoplasma gondii*. A)** Degradation of Rab5b-mAID-HA. Westernblot of Rab5b-mAID HA parasites ± auxin (500 μM) for 30 min and 2 h. Aldolase was used as a loading control. **B)** Representative pictures of the plaque assay of Tir1 and Rab5b-mAID HA ± auxin. **C)** Quantification the plaque size, expressed as percentage of Tir1 – auxin plaque size. White: without auxin, gray: + auxin). **D)**

Quantification of the replication assay after 24 h of parasite replication. Results are expressed as percentage of the total number of vacuole number for each replication stage. **E)** Difference of percentage of each replication stage between Rab5b control and KD parasites. Mean of the difference between the three triplicates. **F)** Invasion assay. Parasites were induced for 2 h prior ± auxin prior invasion. Results expressed as percentage of the number of control parasites. **G)** Representative immunofluorescence images assessing parasite morphology in Rab5b-KD parasites. Markers used: HA for Rab5b, GAP45 for parasite shape, G2-Trx for the apicoplast, RON2-4 for rhoptries, and MIC2, MIC4, MIC8, and AMA1 for micronemes. Three biological replicates were used for all analyses; all P values are 0 ≤ P ≤ 0.001, ***, error bars are standard deviations, and the center measurement of the graph bars is the mean. All scale bars = 1 μm. The data underlying this figure can be found in S1 Data.

In *P. falciparum*, PfVPS45 has been identified as a partner of PfRab5b [32]. In contrast, in *T. gondii*, VPS45 interact with TgStx16, αSNAP, and TgStx6, but not with Rab5b, suggesting differences in the Rab5b-associated complexes between these two apicomplexan parasites [37]. To investigate this further, Rab5b was endogenously tagged with TurboID to perform proximity labeling (S5C Fig) [13]. Upon biotin addition, a rapid shift in the biotinylated protein pattern was observed by IFA and western blot compared to the untreated control (S5D and S5E Fig). A short labeling period of 2 h was chosen for the analysis, as longer incubations (3 and 5 h) resulted in excessive labeling that hindered resolution (S5E and S5F Fig). After isolation of biotinylated proteins by streptavidin pull-down, the samples were sent for mass spectrometry analysis (S5G Fig). Very few proteins were identified as significantly enriched (S5H Fig). Among the top candidates, only a hypothetical v-SNARE protein (TGGT1_219710) could be considered a potential Rab5b partner; the other identified proteins were residents of the apicoplast or mitochondria (S5I Fig). Rab5b itself was not significantly enriched in the biotinylated fraction compared to the control, possibly due to N-terminal myristoylation that embeds the N-terminus in the lipid bilayer, rendering it inaccessible [34,38]. Importantly, VPS45 was not detected in the enriched dataset, consistent with previous findings in *T. gondii* [37]. Further analyses, such as direct pull-down of Rab5b, will be necessary to confirm these results. Taken together, the data further support the notion that Rab5b complexes differ significantly between *T. gondii* and *P. falciparum* in both function and composition.

## MyoF is required for vesicle dynamics and trafficking, but is not critical for endocytosis or recycling

Next, we analyzed a potential molecular motor involved in this process. Together with MyoA, MyoF is the only myosin conserved within the phylum of apicomplexans [39]. Furthermore, MyoF it has been described to be important for several trafficking pathways [40,41], making it a likely candidate to be also involved in endocytosis. We tagged SAG1 with Halo in the previously characterized MyoF-mAID strain that allows auxin-dependent degradation of MyoF [40]. Endocytosis assays were performed as described in Fig 2B with a MNP dye. Depletion of MyoF by the addition of auxin had little impact on the endocytic itself, since no significant differences were observed in the percentage of vacuole where endocytic events were observable (Fig 6A and 6B). Instead, a significant reduction of Endo-SAG1 vesicles inside the cytoplasm of the parasite was observed in the absence of MyoF (Figs 6A, white arrows and 4C) that was correlated with an increase of endocytic vesicles inside the residual body (Figs 6A, yellow arrows and 6D). This suggests that MyoF is not required for endocytosis per se but is important for their transport within the cytosol.

To analyze vesicle trafficking, the movement of the cytoplasmic Endo-SAG1 vesicles was recorded and quantified (Fig 6D and 6E). Three categories were defined: directional movement, diffusive-like movement, and stationery (Fig 6D). Similar to what was observed for the movement of DG [42], depletion of MyoF results in an increased number of stationary vesicles, passing from 6 ± 2% to 25 ± 15% compared to control. This increase of stationary vesicles was correlated with the decrease of SAG1 vesicles with a directed motion, dropping from 42 ± 9% to 18 ± 14%. The percentage of SAG1-vesicles with diffusive-like movement was not strongly impacted (51 ± 1% Ctrl versus 55 ± 5% MyoF-KD). These results suggest that MyoF is an important motor for the trafficking of the SAG1 vesicles, with a strong impairment of the directionality of the movements and its absence leads to an accumulation of the vesicles inside the residual body.

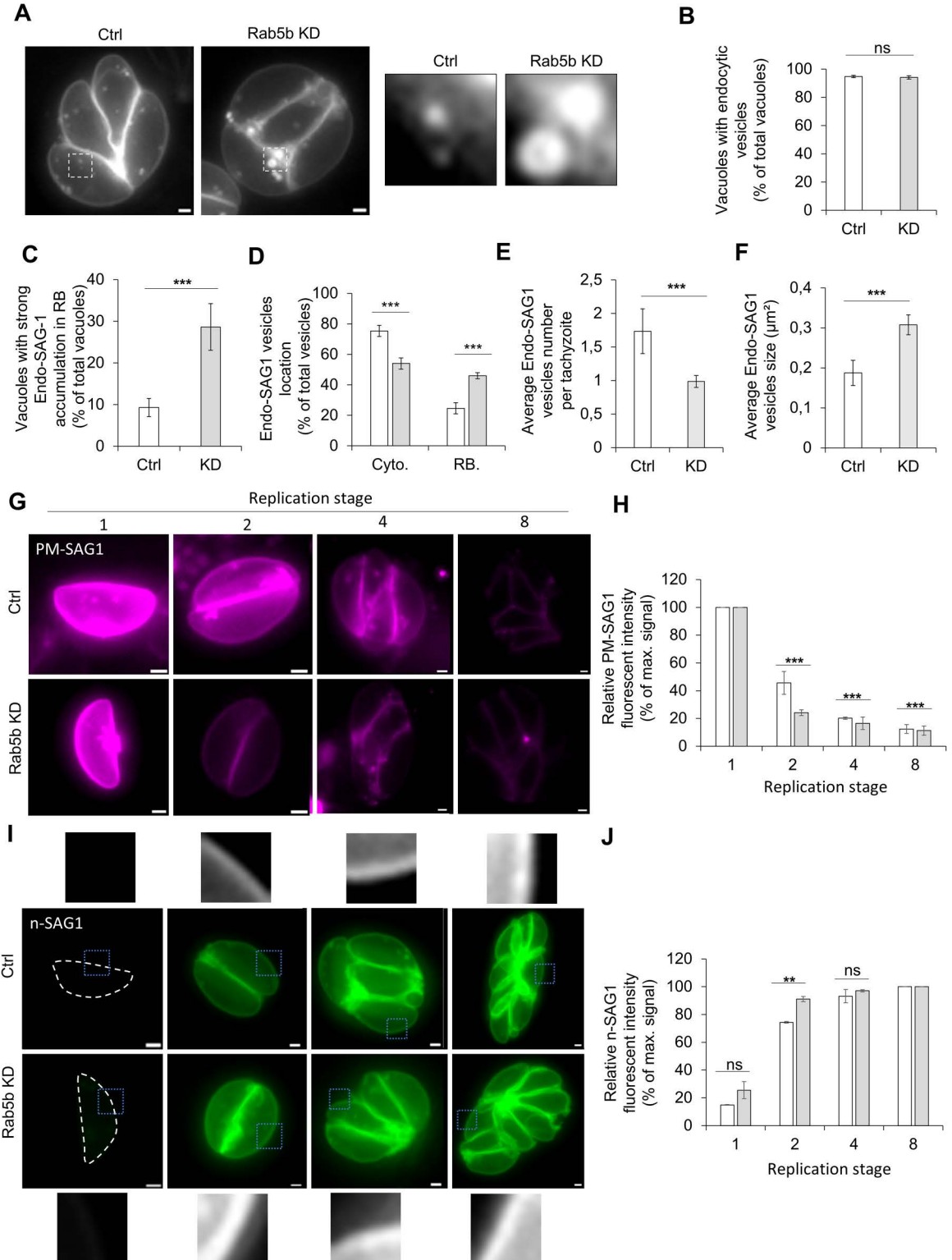

**Fig 5. The plasma membrane recycling is dependent on Rab5b. A)** Endocytosis of the PM. Representative pictures of the endocytosis in the Rab5b-mAID strain ± auxin treatment. Gray: PM-SAG1. The zoom windows of show variation in vesicle size. **B)** Quantification of the percentage of vacuoles containing endocytic vesicles ± auxin. **C)** Quantification of the percentage of vacuoles with strong accumulation of endocytic vesicles in the residual

body. Percentages calculated with the total number of vacuoles. D) Quantification of the distribution of endocytic vesicles between the cytoplasm and the residual body, expressed as a percentage of the total number of vesicles analyzed. E) Average number of Endo-SAG1 vesicles per tachyzoite. For each vacuole, the total number of vesicles was divided by the number of parasites. A total of 25 vacuoles was analyzed per replicate. F) Average Endo-SAG1 vesicles size. A total of 150 vesicles per replicate and condition was analyzed. G) Representative pictures of parasites at different stages of replication (1–8) with PM-SAG1 labeling. H) Quantification of the fluorescence intensity of the plasma membrane PM-SAG1 during replication. I) Representative pictures of parasites at different stages of replication (1–8) with n-SAG1 labeling. Zoom region allow a better visualization of the membrane intensity at the different replicative stage. J) Quantification of the fluorescence intensity of the newly generated n-SAG1 (green) during replication. For all graphs: white bars represent Rab5b-mAID control and gray bars represent Rab5b-KD. Three biological replicates were used for all analyses; all $P$ values are $0 \leq P \leq 0.001$, ***, error bars are standard deviations, and the center measurement of the graph bars is the mean. All scale bars = 1 μm. The data underlying this figure can be found in S1 Data.

As the movement of the vesicles was impaired, we decided to investigate the potential impact on the PM recycling/inheritance process (Fig 6G and 6H). Using a similar protocol as described above for Fig 2, the PM inheritance was analyzed in the presence or absence of auxin. Interestingly, in opposition to what was observed for Rab5B and despite the clear impact on vesicle movement, no significant difference was observed between MyoF Ctrl and KD parasites, with a decrease of the FI at the PM of about 2-fold at each round of replication as observed previously (Fig 2). This suggests that, unlike Rab5b, which affects both trafficking and vesicle size, the absence of MyoF primarily disrupts trafficking without preventing the fusion of endocytic vesicles back to the membrane. Such fusion may therefore still occur, potentially even from within the residual body.

As the inheritance was not impacted, we wondered if differences in the behavior of the sub-populations of SAG1 vesicles (Endo, Int, and n-SAG1 vesicles) could be detected (Figs 6I, 6J, and S7). After induction and triple labeling, the colocalization between all the SAG1 populations was analyzed. A total, 321 Endo-SAG1, 455 Int-SAG1, and 294 n-SAG1 vesicles were counted for the control and 326 Endo-SAG1, 422 Int-SAG1, and 362 n-SAG1 for the KD. Depletion of MyoF did not have a severe impact, as the overall profile remained similar. We noted a reduction in the single (Endo SAG1 only, 34 ± 3% versus 19 ± 12%) and double signal (Endo + Int SAG1, 33 ± 8% versus 18 ± 6%) and an increase of the triple signal (32 ± 8% to 61 ± 7%), suggesting either a slowdown of vesicle transport, which could cause accumulation of the three subpopulation within the same compartment. Interestingly, colocalization between only Endo-SAG1 and n-SAG1 was not observed, supporting our previous data.

Taken together, these results illustrate that the formation of the endocytic vesicles is independent of MyoF, while their subsequent trafficking is impacted by MyoF depletion. Interestingly, this alteration of the trafficking had little impact on the PM inheritance or homeostasis.

## Generation of a PM reservoir

During our analyses, single parasites were regularly observed to form membrane accumulations above the surface of the parasite. These membrane "loops" were formed prior to the onset of replication. When analyzing live replication assays (Fig 7A and S1 Video), we observed the occurrence of these extra membrane loops in 93 ± 5% of the parasites undergoing replication (total $n = 163$ parasites passing from stage 1 to 2, Fig 6B). These structures only appeared transiently. It is formed while the parasite is still in stage one, it enlarges while the daughter parasites form within the mother and is absorbed during their budding. We speculate that these membranous structures are generated as a reservoir to facilitate the budding of the daughter cells from the mother to avoid membrane rupture. We therefore refer to this structure as a plasma membrane reservoir (PMR).

The initial position where we detected the formation of the PMR is variable (S8A Fig). Once formed, the position and the size of PMRs are also variable as they move along the PM of the parasite (S8C Fig). We observed that this structure disappears once the daughters are formed, but reappears when the parasites engage in the next replication stage (S8D Fig). To correlate PMR formation with daughter cell formation, we performed live microscopy with SAG1-Halo IMC-1-YFP

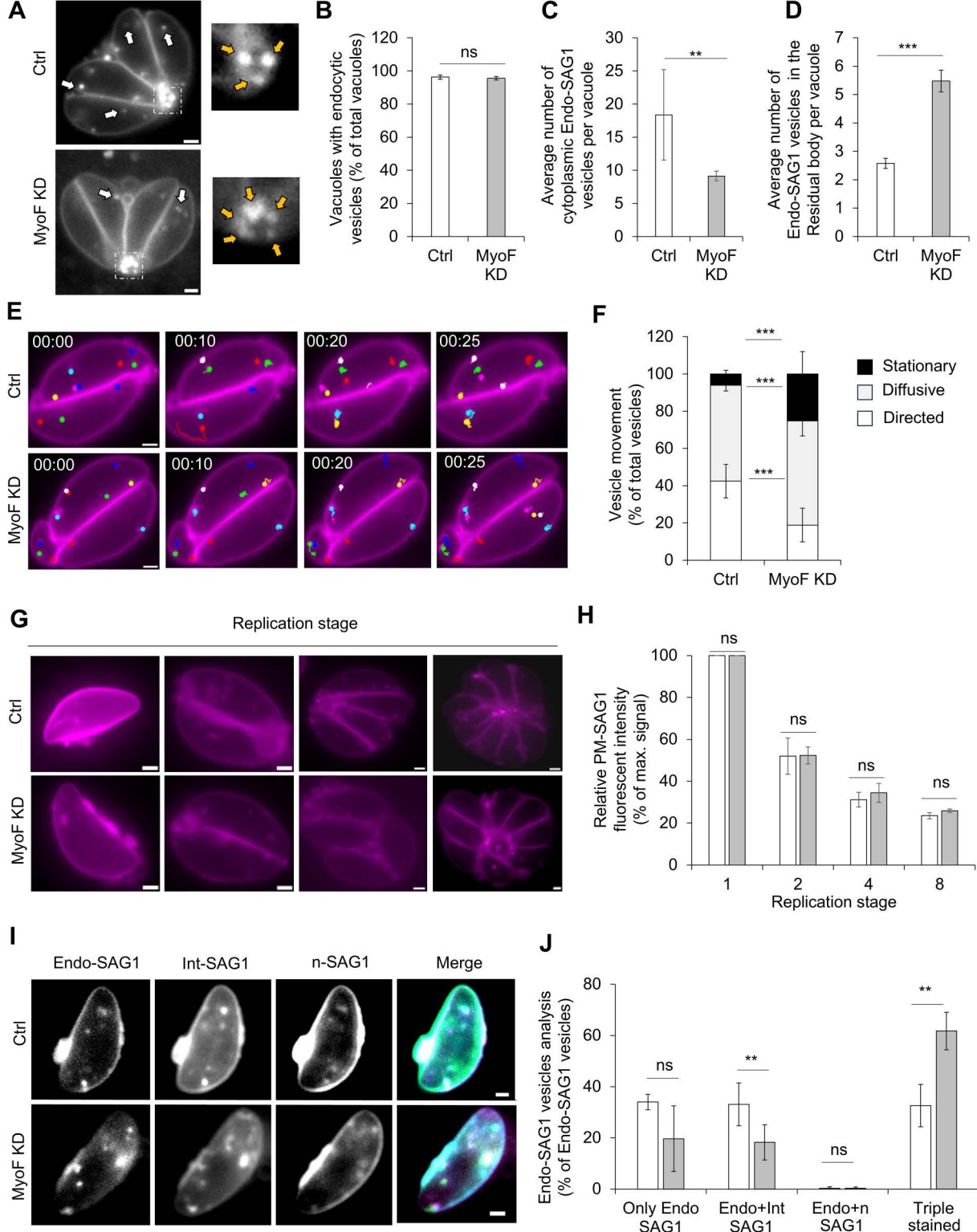

**Fig 6. Only trafficking of the SAG1 vesicles is dependent on MyoF. A)** Endocytosis of the PM. Representative pictures of the endocytosis in the MyoF-mAID strain ± auxin treatment. Gray: PM-SAG1, white arrows: endocytic events in the cytoplasm. The zoom out of the residual body region shows presence of endocytic vesicles which are highlighted with yellow arrows. **B)** Quantification of the percentage of vacuoles where endocytosis

could be observed ± auxin treatment. **C)** Quantification of the average number of endocytic vesicles inside the cytoplasm per vacuole. **D)** Quantification of the average number of endocytic vesicles inside the residual body per vacuole. **E)** SAG1 vesicle trafficking. Time series images of the tracking of Endo-SAG1 vesicles in MyoF-mAID parasites ± auxin. Top series: MyoF ctrl/ − auxin, bottom series: MyoF KD/ + auxin. Time in seconds. Each dot represents a vesicle which was detected. **F)** Quantification of the different Endo-SAG1 vesicle movements recorded for MyoF-mAID parasites ± auxin. Three categories of movements were determined: White: directed movements, Gray: diffusive movement, and Black: stationary. **G)** Inheritance of the PM. Representative pictures of the MyoF-mAID parasites at different stages of replication (from 1 to 8) after labeling of the PM prior invasion. Top series: MyoF ctrl/ − auxin, bottom series: MyoF KD/ + auxin. **H)** Quantification of the fluorescence intensity of PM during replication, illustrated in F. **I)** Representative picture of the triple labeling performed as described in Fig 3. Top series: MyoF ctrl/ − auxin, bottom series: MyoF KD/ + auxin. **J)** Quantification of the colocalization between the different populations of SAG1 vesicles. Experiment was performed in triplicate with 25 parasites analyze per replicate for a total of 321 Endo-SAG1, 455 Int-SAG1, and 294 n-SAG1 vesicles for the control and 326 Endo-SAG1, 422 Int-SAG1, and 362 n-SAG1 for the KD. Endo+Int: vesicles containing Endo-SAG1 and Int-SAG1 only, Endo+n: vesicles containing Endo-SAG1 and n-SAG1 only, Triple: vesicles containing the three populations. Three biological replicates were used for all analyses; for all the quantification graphic comparing control and KD: White: MyoF ctrl/ − auxin, gray: MyoF KD/ + auxin.; all $P$ values are $0 \leq P \leq 0.001$, ***, error bars are standard deviations, and the center measurement of the graph bars is the mean. A one-tailed unpaired Student $t$ test was used for all comparisons with no adjustments. All scale bars = 1 μm. The data underlying this figure can be found in S1 Data.

parasites (Fig 7C and S2 Video). As hypothesized, the appearance of PMR happened before the formation of the daughter's IMC, and it disappeared when the budding of daughters started.

To exclude the possibility that the PMR is an artifact caused by phototoxicity during live imaging, we performed fixed assays after 24 h of replication using parasites expressing SAG1-Halo IMC1-YFP, or SAG1 Halo alone. We confirmed the presence of the PMR in both cases and detected PMR at various replication stages (Fig 7D). As demonstrated in Fig 7A or S8C and S8D Fig, the PMR is very dynamic, and it has to be noted that the presence of the daughter cells does not ensure the presence of the PMR (S8E Fig). In these unsynchronized experiments, we could observe PMR in about 12 ± 3% of the vacuoles (Fig 7E). This low percentage of parasite exhibiting simultaneously a PMR can be explained by the PMR dynamic and transient nature. The simple synchronization of tachyzoites with a pulse invasion allows the visualization of PMRs in 43 ± 7% of the vacuole and up to 85 ± 3% with expansion microscopy (Fig 7E).

To exclude the possibility of the PMR being caused by the expression of SAG1-Halo, we investigated the presence of the PMR on WT parasites. Using transmission electron microscopy, the PMR was also observed (Fig 7F). The presence of PMR in WT parasites was also confirmed using SAG1 antibody and expansion microscopy (Figs 7G and S8B).

Finally, we performed the dual labeling described above (Fig 2D), to differentiate between maternal M-SAG1 and new n-SAG1 to address the composition of the PMR (S8F Fig). We observed both maternal and newly synthesized SAG1 inside the PMR, demonstrating that the production and secretion of de novo PM occurs early, before the first replication.

Taken together, these data demonstrate that the parasite generates, prior to the development of the daughter cells, a transient PMR. This dynamic reservoir is reabsorbed at the end of the replication cycle.

### The balance between endocytosis and exocytosis is critical for the PMR regulation

The dynamic variation in size and position of the PMR during the replication also occurs independently of significant morphological change of the replicating parasite as illustrated in S8C Fig. We therefore speculate that, as for any membranous structure, the PMR is regulated via the balance of exocytosis and endocytosis. We started to determine if the PMR could be associated with the micropore, which was recently described as a critical structure for endocytosis [13], by investigating their potential colocalization (Fig 8A). No significant colocalization was observed between the micropore and PMRs (6 ± 3%, Fig 8B), coherent with the dynamic localization of the PMR (S8 Fig). Yet, the absence of strong colocalization does not exclude the relation between the two structures as observed for the endocytic vesicles and the micropore (Fig 3).

We then examined the impact of the K13-KD which strongly impair the endocytic process on PMR regulation. As described in the previous study, the KD was induced for 24 h with anhydrotetracycline (ATC) prior to labeling and transfer

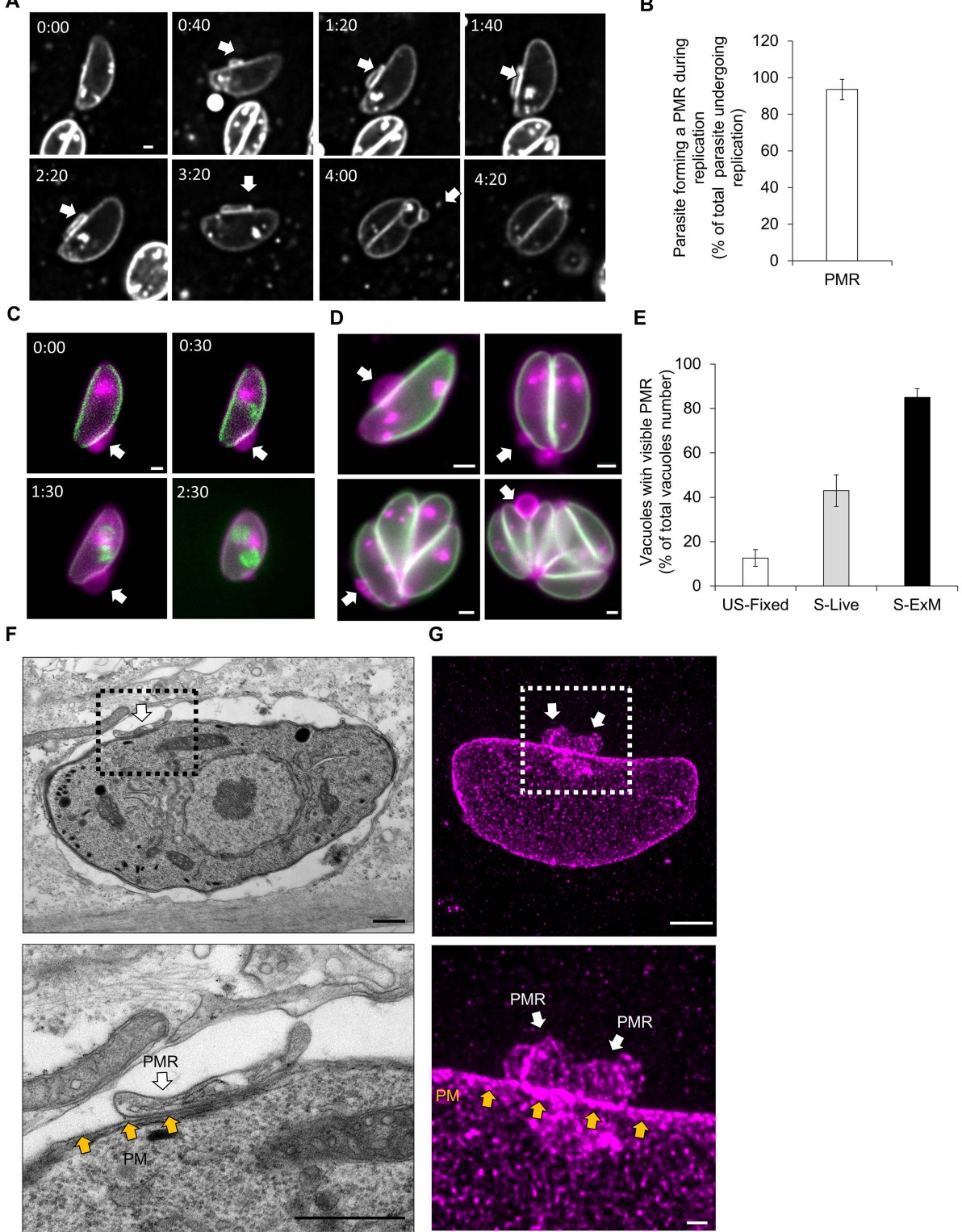

**Fig 7. Discovery of the plasma membrane reservoir (PMR). A)** Time series of the live replication of a SAG1-Halo stained parasite. White arrows indicate the PMR, time in hours and minutes. Scale bar = 1 μm. **B)** Quantification of the percentage of parasites where the formation of the PMR was observed during replication by live microscopy. Expressed as a percentage of total parasites observed undergoing replication. **C)** Time series of the live

replication of a parasite expressing IMC1-YFP (green) and SAG1-Halo (magenta). White arrows indicate the PMR, time in hours and minutes. Scale bar = 1 μm. **D)** Representative pictures of fixed time point microscopy of parasites expressing IMC-YFP (green) and SAG1-Halo (magenta). Formation of the PMR was observed for parasites at different replication stages (1–8). White arrows indicate the PMR. Scale bars = 1 μm. **E)** Quantification of the percentage of vacuoles with PMR. White: unsynchronized parasites after 24-h replication, Gray: synchronized parasites during live replication, black: synchronized parasites imaged with expansion microscopy. **F)** Ultrastructure TEM of a WT parasite (without expression of SAG1-Halo) with a PMR similar to what is observed in A at 01:40. The position of the PMR (white arrows) above the PM (yellow arrows) is highlighted. Scale bars: 500 nm. **G)** U-ExM of WT parasite using SAG1 antibodies. The position of the PMR (white arrows) above the PM (yellow arrows) is highlighted. Scale bars: 5 and 1 μm. Three biological replicates were used for all analyses; error bars are standard deviations, and the center measurement of the graph bars is the mean. The data underlying this figure can be found in S1 Data.

onto the host cell. After 24 h (a total of 48 h of induction), we analyzed parasites ± ATC treatment (Fig 8C). Interestingly, we observed the formation of PMRs in 88 ± 5% of the vacuoles in the K13 KD in opposition to 30 ± 10% in control (Fig 8D). In addition, in the case of the K13-KD the average number of PMR visible per parasite, their average area and total surface also increased (Fig 8E, 8F, and 8G).

Taken together, the data suggest that the disruption of endo-exocytic balance in the absence of a functional micropore leads to an increased generation and dysregulation of PMRs, which are not reabsorbed and accumulate, eventually causing the formation of aberrant membrane structures.

## Discussion

Here, we provide to our knowledge a first characterization of PM dynamics during replication in an apicomplexan parasite. The Halo tag, which we inserted before the GPI anchor of SAG1, allowed us to exploit the full potential of the Halo technology (Figs 2 and 3). Thanks to different dyes with variable properties, we can differentiate between the extracellular and cytoplasmic fractions, as well as between maternal and newly generated material [43]. Using this technology, we are able to analyze endocytosis (Fig 2B and 2C), the secretion of the internal stocks of membrane (Fig 2D), the inheritance of the maternal material, and the secretion of de novo material (Fig 2C). This secretion occurs rapidly after invasion and prior to replication.

To validate the GPI-anchored surface antigen SAG1 as a good proxy for membrane dynamics, FRAP was used, and we confirmed that the tachyzoites within the same vacuole remain connected via the residual body and, in addition to sharing cytosolic content, also share a fluid PM (Fig 1). It has been previously demonstrated that parasites within the same vacuole share their cytoplasmic content using a similar FRAP-based experiment on parasites expressing cytoplasmic GFP. This recovery is abolished in the absence of the residual body [26] or impairment of MyoI/J [27]. As the parasites in the same vacuole are connected by PM bridges (Fig 1), we also expected a recovery for SAG1-Halo. Indeed, we observed such recovery when half of the parasites within a vacuole were bleached, although the required time was longer compared to cytosolic content. While cytosolic GFP was shown to recover within a minute [26,27], we observed a slower recovery requiring 5–10 min. Two hypotheses could explain this difference: (1) the thin PM bridges connecting individual parasites within the PV might not allow the rapid diffusion of membrane proteins, therefore representing a bottleneck compared to the free GFP diffusion inside the cytoplasm, or (2) the protein amount of cytosolic GFP might be more important than the amount PM SAG1-Halo, which lead to a stronger FI signal and facilitate the visualization of the recovery in the case of the GFP. Nevertheless, the rapid recovery observed for SAG1-Halo on a single parasite, irrespective of the size of the bleached area, validates SAG-Halo as a good proxy for membrane dynamics. These experiments demonstrate that *T. gondii* tachyzoites form a syncytium during replication where all the daughters share cytoplasmic and PM content via the residual body. In this respect, the term "residual body" is slightly misleading, since this structure appears to be a key hub for material exchange [43], synchronization of replication and signaling of egress [26,28]. A comparative analysis of membrane dynamics within tissue cysts would be interesting as the absence of cell-cell communication and cytosolic exchange has been demonstrated in the case of bradyzoites [27].

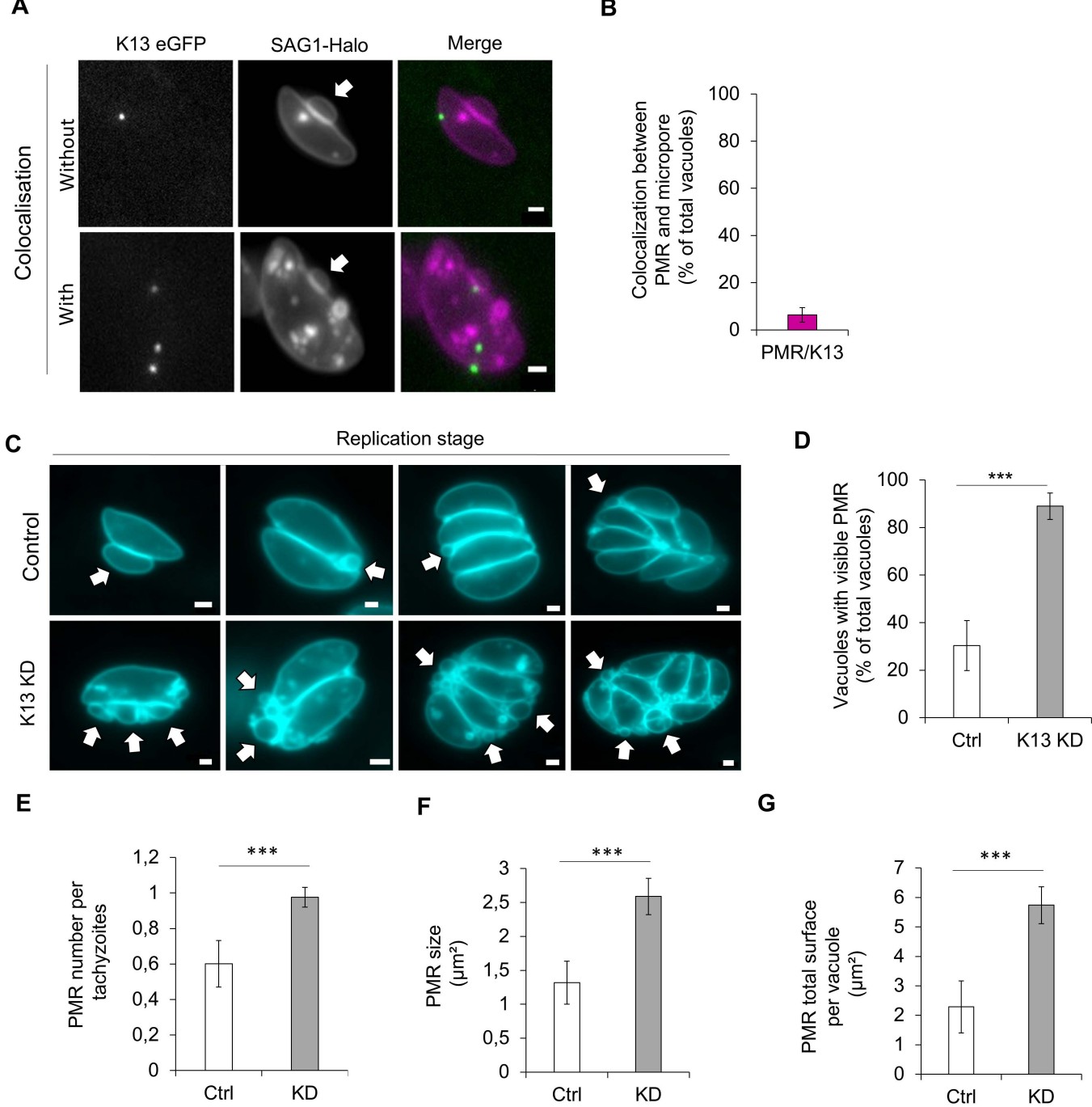

**Fig 8. The endo-exocytic balance is critical for the PMR dynamics. A)** Representative pictures of the colocalisation between the PMR and micropore using K13-eGFP. K13: green, SAG1-Halo: magenta. White arrows indicate the PMRs. **B)** Quantification of the colocalisation between the PMR and the micropore. The result is expressed as a percentage of the total number of vacuoles observed. **C)** Representative pictures of T4S1-TgK13 SAG1-Halo parasites, ±ATC, at different replication stages (from 1 to 8). Cyan: SAG1-Halo Jan 647. White arrows indicate the PMRs. D, E, F, **and** G) Impact of the alteration of the endo-exocytosis balance on the PMR using T4S1-TgK13 SAG1-Halo parasites, ±ATC. White: K13 Ctrl/ −ATC, gray: K13 KD/ +ATC. D) Quantification of the percentage of vacuoles with visible PMR. E) Quantification of the average number of PMR per tachyzoites. F) Quantification of the average PMR size. G) Quantification of the total PMRs surface. Three biological replicates were used for all analyses; all $P$ values are $0 \leq P \leq 0.001$, ***, error bars are standard deviations, and the center measurement of the graph bars is the mean. A one-tailed unpaired Student $t$ test was used for all comparisons with no adjustments. All scale bars = 1 μm. The data underlying this figure can be found in S1 Data.

We analyzed the complete cycle of SAG1-Halo, from de novo synthesis, delivery to the PM, endocytosis, and recycling (Fig 9B). MyoF was identified as an essential motor for the trafficking of endocytic vesicles, but not their formation nor their fusion with the PM (Fig 6). The relatively mild impact of MyoF depletion on membrane inheritance suggests either the involvement of other, potentially redundant, molecular motors [39], or that the vesicles fusion can still occur even in the RB. Consistent with previous results [29], our data indicate that endocytosis and exocytosis pathways intersect, probably near the ELCs where Rab5a is localized, but without fusion of the de novo and recycled vesicles.

While the exact detail of the endocytic pathway requires further investigation, we confirmed the critical role of the small GTPase Rab5B for the endocytosed material. In *P. falciparum*, the proteins PfRab5b, PfRbsn5L, and PfVPS45 form a stable complex that regulates endocytosis and the host cell cytosol uptake (HCCU) [32]. Disruption of any component of the VPS45/Rab5b complex reduced hemoglobin accumulation in parasites by 80% and death of the parasite illustrating the critical role of the PfRab5b/PfRbsn5L/PfVPS45 complex in HCCU. In contrast, Rab5B appears dispensable for the lytic cycle of *T. gondii* but plays a critical role in PM homeostasis through its involvement in endocytic recycling. While DN studies suggested that Rab5B was essential [31], inducible depletion using an mAID system revealed only a mild growth defect, without notable effects on invasion, replication, or organelle integrity. It is well documented that overexpression of DN constructs may elicit phenotypes that are not identical to gene knockout, and in some cases are more severe or fundamentally different [37,44].

Instead, Rab5B loss led to a pronounced alteration in endocytic vesicle dynamics, including reduced vesicle number, increased vesicle size, and abnormal accumulation of endocytosed SAG1 in the residual body. This phenotype was accompanied by a decreased recycling of SAG1 to the PM and a compensatory increase in n-SAG1 synthesis. The loss of signal (Fig 5G and 5H) also suggests that part of the endocytic material might be degraded as accumulation of endocytic material is not observed in all vesicles, degradation which seems to be absent in the case of MyoF depletion (Fig 6G and 6H). These findings indicate that Rab5B functions downstream of endocytosis to promote vesicle recycling and support PM maintenance during replication.

Our data further support the hypothesis that endocytosis at the micropore contributes primarily to PM homeostasis. However, Rab5b could still play a role in host protein uptake despite the relatively mild phenotype observed upon its depletion. One possibility is that the use of nutrient-rich culture media supplies all essential amino acids exogenously, thereby masking any defect in protein uptake. Indeed, differences in culture conditions have been shown to influence the severity of growth phenotypes in mutant parasites [45]. Alternatively, as suggested by the limited uptake of exogenous GFP, protein internalization may not represent a major source of amino acids for *Toxoplasma*. *Plasmodium* and *Toxoplasma* infect very different host cell types. Mature red blood cells lack translational activity due to enucleation and contain only low levels of free amino acids, while *Toxoplasma* invades nucleated, metabolically active cells that are rich in amino acids [46]. Further studies will be necessary to investigate the mechanistic basis for this divergence in Rab5b function between the two apicomplexans.

Finally, *T.* gondii tachyzoites build a reservoir of PM that we refer to as PMR (Figs 9A, 7, and S8). This structure is formed transiently, with neither its position nor its size fixed (Figs 7 and S8). At the end of replication, the PMR is reabsorbed and can be translocated to the posterior end of the parasite, where it becomes indistinguishable from the residual body [23,47]. At this stage, the exact function of the PMR remains unclear. Taking our results into consideration, we hypothesize that PMR formation could be: (a) a process that anticipates the future expansion of the PM required during daughter cell budding; (b) an excess of membrane to support the intense endocytic activity occurring during replication; (c) a membrane reserve that facilitates the future connection of daughter cells after their emergence; or (d) a combination of these hypotheses. Nevertheless, all the parasites undergoing replication formed this PMR (Fig 7A and 7B).

Membrane reservoirs similar to the PMR have been reported in various organisms during cell division, ranging from unicellular organisms such as *Dictyostelium* to human cells [48,49]. Despite differences in the replication processes, analogous membrane accumulations have been observed in *Plasmodium*, where they facilitate parasite development in both

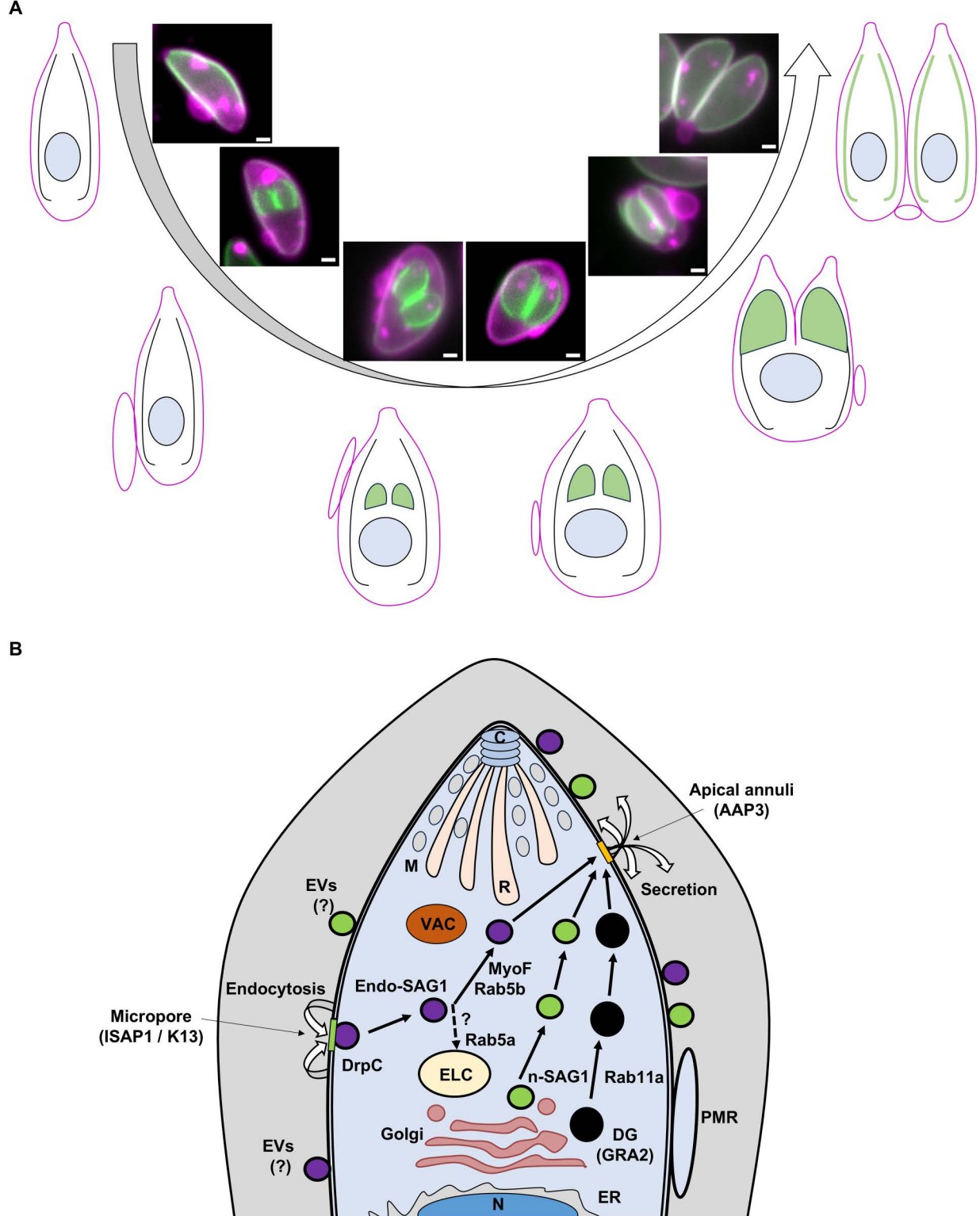

**Fig 9. Schematic summary. A)** Formation and Dynamics of the PMR. The PMR forms prior to the development of the daughter cells' IMC. Its position and size vary during replication, and it is reabsorbed at the conclusion of daughter cell budding. We propose that the PMR facilitates the expansion of the PM surface during replication. All scale bars represent 1 µm. While the PMR is not always visible due to its transient nature, virtually all the parasites

undergoing replication are forming a PMR. **B)** Summary of the colocalisation analysis of endocytic vesicles or the Endo-SAG1 cycle during replication. Endocytosis occurs at the micropore in a rapid process, where a small number of vesicles can colocalise with ISAP1 or DrpC. The Endo-SAG1 vesicles are subsequently trafficked via MyoF, with Rab5b appearing as a potential trafficking factor. Some colocalisation with Rab5a, which is crucial for microneme function, suggests potential interactions with the ELC or TGN. The trafficking of Endo-SAG1 is independent of the n-SAG1 and DG (Rab11a and Gra2) pathways. The limited colocalisation with AAP3 may indicate that secretion occurs at the apical annuli. SAG1 vesicles observed within the parasitophorous vacuoles might represent extra vesicles or EVs.

liver and mosquito stages [50]. The generation of PMR during cell division seems to be a conserved mechanism across evolution. Interestingly, in all these studies, the regulation of these PMRs was dependent on endocytosis and exocytosis [48,49,51]. Here, we demonstrate that endocytosis and exocytosis are similarly linked to PMR regulation in *Toxoplasma* (Fig 8). While we have highlighted the importance of the endo-exocytic balance for PMR regulation, several questions remain unanswered. How is PMR generation controlled, from secretion to positioning? Does the lipid composition at the junctions change to enable the necessary membrane curvatures? Is PMR absorption coupled with nutrient uptake? Further investigations are needed to address these fundamental questions.

## Materials and methods

### Growth and generation of transgenic *T. gondii*

*T. gondii* tachyzoites from the strain RH and derived strains, including RH Δku80/TATi and RH Δku80/Tir1, were maintained at 37 °C with 5% $CO_2$ growing in human foreskin fibroblasts (HFFs; ATCC, SCRC 1041) cultured in Dulbecco's Modified Eagle Medium (DMEM Sigma, D6546) supplemented with 10% fetal bovine serum (FBS BioSell FBS.US.0500), 4 mM L-Glutamate (Sigma, G7513), and 20 µg/mL gentamycin (Sigma, G1397) as previously described [16].

### Generation of transgenic parasites

New strains were generated via the use of CRISPR/Cas9 as previously described [52]. Guide RNAs targeting the regions of interest were designed using EuPaGDT [53]. The sequences of all gRNAs and primers can be found in S2 Data. Briefly, gRNA oligos were annealed, ligated into the Cas9-YFP vector and verified by sequencing (Eurofins Genomics). Reparation templates DNA were generated by amplifying the Halo or YFP with 50 bp homology arms to the gene of interest by PCR using Q5 High-Fidelity DNA Polymerase (New England BioLabs). The repair template was purified using a PCR purification kit (Blirt, EM26.1). Tachyzoites were pooled with the repair template PCR and 10 µg of the corresponding Cas9 vector. Parasites were then transfected with Amaxa 4D-Nucleofector system (Lonza AAF-1003X). The transfected parasites were allowed to invade fresh HFFs and replicate for 48 h. After manual egress and filtration through a 3 µm filter, the parasites were enriched for those expressing Cas9-YFP using FACS (FACSARIA III, BD Biosciences) and sorted into 96-well plates. Proper integration of repair templates was confirmed via PCR and sequencing (Eurofins Genomics).

### SAG1 Halo labeling strategies

**Labeling of the PM-SAG1 and visualization of the endocytosis/Endo-SAG1.** Fresh tachyzoites expressing Sag1-Halo were mechanically egressed, filtered (3 µm), and resuspended in cold DMEM media containing Halo MNP dye Alexa 660 or 488 dye (Halo-Alexa Fluor-488/660 1:1,000 Promega) for 1 h. Parasites were spined at 2,500 rpm for 5 min and media was exchanged, this was repeated three times, to remove any unbound ligand. Parasites were then transferred to Ibidi live cell dishes covered with HFF for overnight replication before imaging.

**Labeling of the maternal M-SAG1 and visualization of the n-SAG1.** Fresh tachyzoites expressing Sag1-Halo were mechanically egressed, filtered (3 µm), and resuspended in cold DMEM media containing Halo membrane-permeable dye, Janilla 646 (Janeli Fluor-646, 1:1000, Promega) for 1 h. Parasites were spined at 2,500 rpm for 5 min and media exchanged, three times, to remove any unbound ligand. Parasites were then transferred to Ibidi live cell dishes covered

with HFF for overnight replication. After replication, a new labeling with media containing Halo membrane-permeable dye, Jan 549, (Janelia Fluor-549, 1:1,000, Promega) was performed for 1 h at 37 °C. After three washes to remove any unbound dye left, parasites were imaged.

**Labeling of the Int-SAG1 and tracking of the distribution to the daughter cells.** Fresh tachyzoites expressing Sag1-Halo were mechanically egressed, filtered (3 µm), and resuspended in cold DMEM media containing Halo MNP dye Alexa 660 or 488 dye (Halo-Alexa Fluor-488/660 1:1,000 Promega) for 1 h. Parasites were spined at 2,500 rpm for 5 min and media exchanged, three times, to remove any unbound ligand. Parasites were then resuspended in cold DMEM media containing Halo membrane-permeable dye Janila 549 or 646 (Janila Fluor-549,646 1:1,000, Promega) for 1 h. Parasites were spined at 2,500 rpm for 5 min and media exchanged, three times, to remove any unbound ligand. Parasites were then transferred to Ibidi live cell dishes covered with HFF for overnight replication before imaging.

**Triple labeling.** Fresh tachyzoites expressing Sag1-Halo were mechanically egressed, filtered (3 µm), and resuspended in cold DMEM media containing Halo MNP dye Alexa 488 (Halo-Alexa Fluor-488 1:1,000 Promega) for 1 h. Parasites were spined at 2,500 rpm for 5 min and media exchanged, three times, to remove any unbound ligand. Parasites were then resuspended in cold DMEM media containing Halo membrane-permeable dye 646 (Janelia Fluor 646 1:1,000, Promega) for 1 h. Parasites were spined at 2,500 rpm for 5 min and media exchanged, three times, to remove any unbound ligand. Parasites were then transferred to Ibidi live cell dishes covered with HFF for overnight replication. After replication, a new labeling with media containing Halo membrane-permeable dye, Jan 549, (Janelia Fluor-549, 1:1,000, Promega) was performed for 1 h at 37 °C. After three washes to remove any unbound dye left, parasites were imaged.

## Western blots

For western blot analysis, $1.5 \times 10^8$ parasites were mechanically egressed, filtered (3 µm), and washed three times with cold PBS. Pellets were resuspended in 25 µl RIPA buffer with Triton X-100 and EDTA-free protease inhibitor cocktail (Invitrogen, Thermo Fisher Scientific, Waltham, MA, USA) (RIPA: 0.5% sodium deoxycholate, 150 mM NaCl, 1 mM EDTA, 0.1% SDS, 50 mM Tris-HCl pH 8.0, 1% Triton X-100), and incubated on ice for 5 min. Lysates were centrifuged at maximum speed for 3 min at 4 °C. The supernatant was transferred to a new tube and mixed 1:1 with 4× Laemmli loading buffer containing 100 mM DTT, followed by boiling at 95 °C for 10 min.

Each sample was loaded onto 4%–20% Mini-PROTEAN TGX precast gels (Bio-Rad, Hercules, CA, USA) before transfer onto a nitrocellulose membrane. Membranes were blocked with 5% skimmed milk in PBS for 1 h at room temperature before incubation with primary antibodies overnight at 4 °C. Primary antibodies for detection of Rab5b-mAID-HA expression were a rat anti-HA High Affinity monoclonal antibody (clone 3F10, 1:500; Roche Diagnostics GmbH, Mannheim, Germany; cat. #1187431001) and a rabbit anti-Aldolase antibody (1:10,000; kindly provided by David Sibley).

After three washes, membranes were incubated for 1 h at room temperature with secondary antibodies (IRDye 680RD Goat anti-Rat IgG, 1:10,000, and IRDye 800CW Goat anti-Rabbit IgG, 1:10,000; both from LI-COR Biosciences, Lincoln, NE, USA) diluted in blocking buffer. For Rab5b-Turbo, Streptavidin-800CW was used as sole reagent (IRDye 800CW Goat anti-Streptavidin IgG, 1:10,000, LI-COR Biosciences, Lincoln, NE, USA). Following incubation and three washes, imaging was performed using the LI-COR Odyssey CLx imaging system. Western blot images were processed using LI-COR Image Studio software (version 5.2), with only linear adjustments applied uniformly across the entire blot.

## TurboID

Freshly egressed tachyzoites expressing Rab5b-TurboID and RHΔKu80 parasites were incubated with biotin (150 µM) for 2 h. Parasites were mechanically egressed, filtered (3 µm), and counted to obtain $1 \times 10^8$ parasites per sample. Parasite pellets were thawed on ice and lysed in 950 µl of RIPA buffer (0.5% sodium deoxycholate, 150 mM NaCl, 1 mM EDTA, 0.1% SDS, 50 mM Tris-HCl, pH 8.0, 1% Triton X-100) supplemented with an EDTA-free protease inhibitor cocktail (PI) (Invitrogen, Thermo Fisher Scientific, Waltham, MA, USA). Lysates were incubated on ice for 30 min, in parallel,

Dynabeads M-280 Streptavidin (Invitrogen, Thermo Fisher Scientific, Waltham, MA, USA) were prepared according to the manufacturer's instructions and washed three times with cold PBS using a magnetic stand. The prepared lysate was added to the washed beads and incubated for 1 h at 4 °C with gentle rotation. After the collection of a flow-through sample, beads were washed five times with 1 ml RIPA buffer containing PI, then three times with 1 ml of 150 mM Tris buffer (pH 8.0) without PI. The supernatant was removed, and the beads were stored at −80 °C until mass spectrometry analysis.

## Mass spectrometry analysis

Beads were incubated with 10 ng/µL trypsin in 1M urea 50 mM $NH_4HCO_3$ for 30 min, washed with 50 mM $NH_4HCO_3$, and the supernatant digested overnight in presence of 1 mM DTT. Digested peptides were alkylated and desalted prior to LC–MS analysis. For LC–MS/MS purposes, 5 µl of desalted peptides were injected in an Ultimate 3000 RSLCnano system (Thermo), separated in a 25 cm analytical column (75 µm ID, 1.6 µm C18, Aurora-IonOpticks) with a 50-min gradient from 2% to 35% acetonitrile in 0.1% formic acid. The effluent from the HPLC was directly electrosprayed into an Orbitrap Exploris 480 (Thermo) operated in data-dependent mode to automatically switch between full scan MS and MS/MS acquisition. Survey full scan MS spectra (m/z 350–1,400) were acquired with resolution 60,000 at $m/z$ 400 (max IT 40 ms, AGC target of $3 \times 10^6$). The 15 most intense peptide ions with charge states between 2 and 6 were sequentially isolated to a target value of $2 \times 10^5$, fragmented at 30% normalized collision energy, and acquired with resolution 15,000 at $m/z$ 400 (max IT 25 ms). Typical mass spectrometric conditions were: spray voltage, 1.5 kV; no sheath and auxiliary gas flow; heated capillary temperature, 275 °C; ion selection threshold, $5 \times 10^3$ counts; dynamic exclusion, 20 s).

The MSFragger 4.3-DDA+ workflow [54] from FragPipe 23.0 [55] was used to identify and quantify the proteins from the database Uniprot_UP00000152_Toxoplasmagondii _Me49_downloaded on 24/10/2022 containing decoy entries. The workflow included Percolator 3.7.1 for additional rescoring and posterior error probability calculation [56], ProteinProphet [57] for protein inference, Philosopher 5.1.1 [58] for false discovery rate (FDR) filtering, and IonQuant to quantify the proteins [59]. FragPipe quantified proteins were $log_2$ transformed, missing values imputed randomly from a normal distribution (width: 0.3, downshift:1.8.) and an adjusted $t$ test applied to identify the differentially abundant proteins upon biotinylation in Perseus 2.0.11.0 [60]

## Phenotypic assays

**Induction of KD.** *MyoF:* As previously described for MyoF mAID parasites [40], MyoF-mAID SAG1-Halo parasites were induced ± auxin for 4 h prior to labeling and experiments.

*K13:* As previously described [13], T4-S1 K13 SAG1-Halo parasites were induced 24 h ± ATC prior labeling and experiments

*Rab5b:* For all experiments, Rab5b parasites were induced ± auxin for 2 h prior labeling and experiments.

**Fluorescence recovery after photobleaching (FRAP).** For analysis of the membrane dynamics, SAG1 Halo parasites were inoculated on Ibidi live cell dishes covered with HFF and labeled with Halo-Alexa 488 (MNP) for 1 h and washed prior to imaging. (Halo-Alexa 488 1:1,000, Promega). Photobleaching on the Leica-DMI8, with a FRAP unit, was recorded at objective 100× on single parasites for a complete period of 50 s. (10 s before bleach, 1 s 100% laser power (bleaching event), 39 s recovery). The imaging area was selected to allow the presence of at least another invaded tachyzoite to determine the photobleaching induced by the recording. A total of 25 parasites per replicate were recorded. The experiment was performed in triplicates. Analysis was performed via Fiji, values were normalized using the control parasites and then converted to percentage using 10 s recorded before the bleaching as reference (100%).

For analysis of the recovery of bigger vacuoles, the parasites were labeled similarly after 24 h of replication. The photobleaching was performed on a Leica SP8X STED 3D DLS. Analysis was performed via Fiji.

**Endocytosis assay.** For evaluation of endocytosis, parasites were labeled as described above for PM/Endo-SAG1 with an MNP dye and transferred on Ibidi live cell dishes covered with HFF for overnight. At least 10 fields of view were

recorded for a total of about 100 vacuoles per replicate. The experiment was performed with three independent biological triplicates. For each replicate, we calculate the percentage of vacuoles where endocytic vesicles could be observed. The values are then expressed as the mean values of the three independent experiments ± SD.

**Fluorescence intensity across the replication.** Parasites were labeled as described above in SAG1 Halo labeling strategies and let to replicate for 24 h on Ibidi live cell dishes covered with HFF. About 15 fields of view were images and about 25 vacuoles of 1, 2, 4, and 8 stages were analyzed. For all experiments, maximal FI was measured at the PM. The selected region excluded regions where membrane overlaps were observed with other parasites (from the same vacuole or nearby), or with obvious signal differences in comparison with the rest of the PM. For each replicate, the maximal value recorded among the different stages was set up as the maximum (100%) and used to calculate the percentage of the value measured in the other stages of replication. The experiment was performed with three independent biological triplicates. The values are then expressed as the mean values of the three independent experiments ± SD.

**Colocalisation with Cb-SNAP.** For colocalization between SAG1 and F-actin, parasites were labeled with the SNAP membrane-permeable dye SNAP 647 (Halo-Alexa Fluor660 1:1000 NEB) for 1 h. Parasites were spined at 2,500 rpm for 5 min and media was exchanged. We repeated this step three times, to remove any unbound ligand. Parasites were then labeled with the Halo membrane permeable dye Janila 594 (Halo-Janelia Fluor594 1:1,000 Promega) for 1 h. Parasites were spined at 2,500 rpm for 5 min and media was exchanged. We repeated this step three times, to remove any unbound ligand. Parasites were then transferred to Ibidi live cell dishes covered with HFF for overnight replication. Parasites were then fixed with 4% paraformaldehyde (PFA) and imaged.

For detection of GAP45, the parasites were labeled as described above and then permeabilized with PBS containing 2% BSA + 0,2% Triton X-100 for 30 min. GAP45 was labeled using antibodies against GAP45 (1/5,000) for 1 h. After washes, primary antibodies were detected using a secondary anti-rabbit conjugated with Alexa-350.

For quantification of the colocalisation, 25 vacuoles with a visible F-actin network were selected and analyzed for the colocalisation between the network and the PM through Z-Stack imaging. The experiment was performed with three independent biological triplicates. The values are then expressed as the mean values of the three independent experiments ± SD. Images were analyzed via Fiji.

**Colocalization assay with YFP markers.** For all colocalization assays, all parasite lines were labeled with the Halo MNP dye Alexa 660 (Halo-Alexa Fluor660 1:1,000 Promega) for 1 h. Parasites were spined at 2,500 rpm for 5 min and media was exchanged. We repeated this step three times, to remove any unbound ligand. Parasites were then transferred to Ibidi live cell dishes covered with HFF for overnight replication. Parasites were then fixed with 4% PFA and imaged. For the percentage of vacuole colocalization, 10 fields of view were imaged with Z-stack for a total of at least 100 vacuoles. For the percentage of vesicle colocalization, about 25 vacuoles where colocalization was observed were analyzed for a total of at least 100 endocytic vesicles per replicate. Colocalization with the YFP markers was calculated. The experiment was performed with three independent biological triplicates. The values are then expressed as the mean values of the three independent experiments ± SD. Videos were analyzed via Fiji.

**Endo-SAG1, Int-SAG1, and n-SAG1 colocalization.** For the analysis of the Endo-SAG1 versus n-SAG1 vesicle colocalization, triple labeling was performed as described above. Parasites were transferred on Ibidi live cell dishes covered with HFF for overnight replication. Parasites were then fixed with 4% PFA and imaged with Z-stack. The analysis focused on stage one parasites to avoid the obtention of too much de novo material in the recycling steps. Per replicate, at least 25 parasites were analyzed for a total of about 100 Endo-SAG1 vesicles. The different colocalization categories (Endo-SAG1 alone, Endo-Int SAG1, Endo-new SAG1, and triple labeling) were quantified independently and expressed as a percentage of the vesicles analyzed. Because the Endo-Int/Endo-new colocalization also included signal from triple labeling, the percentage of triple labeling was subtracted from both dual colocalizations to obtain the specific values for each. The experiment was performed with three independent biological triplicates. The values are then expressed as the mean values of the three independent experiments ± SD. Videos were analyzed via Fiji.

 

**Vesicles tracking.** For tacking the Endo-SAG1 vesicles, parasites were labeled as described above with the Halo MNP dye Alexa 488 (nonpermeable Halo-Alexa Fluor-488 1:1,000 Promega). Parasites were transferred to Ibidi live cell dishes covered with HFF for overnight replication. The induction of the MyoF-KD was performed as described above and maintained during replication. Live cell imaging was then performed for 30 s taking 1 fps with the Leica DMI8 100×. At least 100 vesicles were analyzed per replicate. Vesicles were manually analyzed for movement with Fiji, and divided into three categories: directed movement, diffusive movement, or stationary as previously described [42].

**Plaque assay.** Freshly egressed parasites were filtered through a 3 μm filter and pelleted by centrifugation at 2,500 × g for 5 min at 4 °C. Parasites were then resuspended in medium, and $1 \times 10^3$ parasites per well were seeded onto confluent HFF monolayers in 6-well plates with or without auxin (500 μM) and allowed to replicate for 7 days. After incubation, wells were washed once with PBS and fixed with ice-cold 100% methanol for 20 min.

Cells were stained using a Hemacolor staining kit (Merck, Sigma-Aldrich, Darmstadt, Germany). After a final wash with distilled water, plates were air-dried completely. Plaques were imaged and plaque size quantified using Fiji/ImageJ software.

**Invasion and replication assay.** Invasion and replication assays were performed as previously described [13]. Parasites were mechanically egressed, filtered through a 3 μm filter, and pretreated with or without auxin (500 μM) for 2 h. For the replication assay, $1 \times 10^6$ parasites were added to HFF monolayers grown on coverslips in 24-well plates and allowed to invade for 1 h at 37 °C. After three washes to remove extracellular parasites, fresh culture medium containing ± auxin was added. Parasites were allowed to replicate for 24 h before fixation with 4% PFA for 20 min at room temperature, followed by immunofluorescence analysis.

To assess replication, parasitophorous vacuoles were counted and classified based on their replication stage (1, 2, 4, 8, or 16 parasites per vacuole). Results are expressed as the percentage of the total number of vacuoles counted. To evaluate the difference between control and Rab5b KD, the percentage of each stage in the KD condition was subtracted from the corresponding percentage in the control condition. The average difference was then calculated from three independent triplicates. Invasion was quantified by counting the number of vacuoles per field of view (FOV).

**Immunofluorescence analysis.** Parasites were induced ± auxin (500 μM) for 2 h prior to transfer onto host cells. After 24 h of replication ± auxin, parasites were fixed with 4% PFA and permeabilized in blocking buffer (0.2% Triton X-100, 2% bovine serum albumin [BSA] in PBS) for 30 min at room temperature. Antibodies were diluted in the same blocking buffer.

The following primary antibodies were used: α-HA (1:500; Roche, cat. #11874310001) for detection of Rab5b-mAID-HA; α-GAP45 (1:5000; kindly provided by Dominique Soldati-Favre) for the parasite structure; α-G2-Trx (1:500; kindly provided by Lilach Sheiner) for the apicoplast; α-RON2–4 (1:500; kindly provided by Maryse Lebrun) for the rhoptries; and α-MIC2 (1:500; kindly provided by Vern Carruthers), α-MIC4 (1:3,000; kindly provided by Dominique Soldati-Favre), α-MIC8 (1:500; kindly provided by Markus Meissner), and α-AMA1 (1:500; kindly provided by Gary Ward) for the micronemes.

Labeling with primary antibodies was performed for 1 h, followed by three washes and a 1-h incubation with Alexa Fluor-conjugated secondary antibodies (Alexa Fluor 350, 488, or 647; 1:1,000; Invitrogen, Thermo Fisher Scientific, Waltham, MA, USA). For the detection of biotinylated proteins, Alexa Fluor 594-conjugated streptavidin (1:1,000; Invitrogen, Thermo Fisher Scientific, Waltham, MA, USA) was used directly without a primary antibody.

Coverslips were mounted using ProLong Gold Antifade Mountant (Invitrogen, Thermo Fisher Scientific, Waltham, MA, USA) and imaged using a Leica DMI8 microscope.

**Quantification of the PMR formation.** For experiments of quantification of the PMR, the different SAG1 Halo parasite lines were labeled with the Halo MNP dye Alexa 488/660 (Halo-Alexa Fluor-488/660 1:1,000 Promega) or the Halo membrane-permeable dye Janila 594/646 (Janila Fluor-549/646 1:1,000, Promega) or a combination of both (see labeling strategies) for 1 h as described above. Parasites were transferred to Ibidi live cell dishes covered with HFF for overnight replication. About 10 fields of view were imaged with Z-stack for a total of at least 100 vacuoles. For

quantification of the PMR, we counted the number of parasites showing membrane loops or accumulation above the PM. The results are expressed as a percentage of the total parasite population imaged. The experiment was performed with three independent biological triplicates. The values are then expressed as the mean values of the three independent experiments ± SD. Images were analyzed via Fiji.

**Live replication assay.** SAG1-Halo and SAG1-Halo IMC1-YFP parasites were labeled with the Halo membrane-permeable dye Janila 646 (Janila Fluor-646 1:1,000, Promega) for 1 h. After three washes to remove any excess dye, the parasites were transferred to Ibidi live cell dishes covered with HFF. Parasites were allowed to invade for 1–2 h to obtain enough invasion events to reach an average of 10 parasites per FOV. After the invasion, the excess of parasites was removed by three washes. Live cell dishes were then transferred to the Leica-DMI8, heated at 37 °C under a chamber containing 5% $CO_2$. Laser power and exposure were adjusted to the lowest values allowing reliable imaging. Imaged were taken every 20–30 min to follow the PM behavior during replication for 12 h. The experiment was performed with three independent biological triplicates. The values are then expressed as the mean values of the three independent experiments ± SD. Videos were analyzed via Fiji.

**Visualization of PMR by ultrastructure expansion microscopy (U-ExM).** Freshly egressed RH tachyzoites were allowed to infect HFFs in DMEM, supplemented with 5% FBS. After 1 h of infection cells were washed to remove the extracellular tachyzoites and preserve only the intracellular parasites and allowed to replicate for 3 h.

U-ExM was performed via published protocols for *T. gondii* [42]. In brief, coverslips coated with the HFF cells infected with the parasites were typically not fixed before immersion in a 0.7% formaldehyde/1% acrylamide (FA/AA) solution for 5 h at 37 °C. The coverslips were then overlaid with a monomer mixture (19% sodium acrylate/10% acrylamide/0.1% bis-acrylamide) supplemented with 0.5% ammonium persulfate (APS) and 0.5% tetramethylethylenediamine (TEMED) to initiate gel polymerization. After a 30-min incubation at 37 °C, the fully polymerized gels were detached and subjected to a 30-min incubation at 95 °C in denaturation buffer (200 mM SDS, 200 mM NaCl, 50 mM Tris, pH 9). The gels were expanded overnight in successive water baths. The following day, the gels were shrunk in PBS, sectioned, and agitated for 3 h at 37 °C in a solution of appropriate primary antibodies diluted in freshly prepared 2% PBS-BSA. After three 10-min washes in 0.1% PBS-Tween, the gels were incubated for 3 h at 37 °C (agitated and in the dark) with the appropriate secondary antibodies. After three additional 10-min washes in 0.1% PBS-Tween, the gels were expanded overnight in water. Imaging was performed using a Leica TCS SP8 inverted microscope equipped with an HC PL Apo 100×/1.40 Oil CS2 objective and HyD detectors. Z-stack images were acquired via Leica LAS X software and deconvolved with the built-in "Lightning" mode or manually using Huygens deconvolution software. Image processing was conducted via ImageJ software, and maximum projections were generated for publication.

## Imaging

**Widefield microscopy.** Unless stated otherwise, all images were acquired on a Leica-DMI8, objective 100× with the LasX software (v3.7.4). Fiji (v1.53c) was used to analyze the picture and all counts were made manually. LasX software (v.) from Leica was used to obtain parasite imaging data and all images and movies were processed using Fiji (ImageJ) software v Image Processing Software [61].

**Ultrastructure TEM.** For looking at the PM connection of parasites from the same vacuoles, and PMR, RH parasites were transferred to Ibidi μ-dishes previously seeded with HFF cells. After 24 h of replication, the parasites were fixed with 2.5% glutaraldehyde in 0.1 M phosphate buffer pH 7.4. The parasites were washed three times at room temperature with PBS (137 mM NaCl2, 2.7 mM KCl, 10 mM Na2HPO4, 1.8 mM KH2PO4, pH 7.4) and post-fixed with 1% (w/v) osmium tetroxide for 1 h. After washing with PBS and water, the samples were stained *en bloc* with 1% (w/v) uranyl acetate in 20% (v/v) acetone for 30 min. Samples were dehydrated in a series of graded acetone and embedded in Epon 812 substitute resin. Ultrathin sections (thickness, 60 nm) were cut using a diamond knife on a Reichert Ultracut-E ultramicrotome. Sections were mounted on collodium-coated copper grids, post-stained with lead citrate (80 mM, pH 13) and examined

with an EM 912 transmission electron microscope (Zeiss, Oberkochen, Germany) equipped with an integrated OMEGA energy filter operated in the zero-loss mode at 80 kV. Images were acquired using a 2k × 2k slow-scan CCD camera (Tröndle Restlichtverstärkersysteme, Moorenweis, Germany).

**Software.** Fiji (FIJI ImageJ v1.54f) was used to analyze the picture [61] and all counts were made manually.

**Data analysis.** All data were plotted using Microsoft Excel

**Statistical analysis.** For statistical relevance, one-tailed unpaired Student $t$ tests were performed.

## Supporting information

**S1 Fig. Comparison between Nile Red and SAG1-Halo labeling.** Representative picture of labeling with Nile Red and SAG1-Halo with Alexa 488. Scale bar 1 μm.
(TIF)

**S2 Fig. F-actin filaments are inside the parasite symposium and covered by the plasma membrane. (A)** The plasma membrane connection allowed the passage of F-actin from one parasite to the other. SAG1-Halo was visualized using a membrane-permeable dye Jan-647, the CB-snap with a membrane-permeable dye SNAP-594 and GAP45 using antibodies. The F-actin is located where SAG1-Halo is present, without any GAP45 staining. Scale bars 1 μm. **(B)** PM association with the F-actin network on larger vacuoles. SAG1-Halo was visualized using a membrane-permeable dye Jan-647 and the F-actin network with the expression of Cb-SNAP. Scale bars 1 μm. **(C)** Quantification of colocalisation between SAG1 Halo and Cb-SNAP. The percentage of vacuole where colocalisation between SAG1-Halo and Cb-SNAP was calculated. For each replicate, 25 vacuoles with visible F-actin network were selected and then analyzed for the presence of SAG1-Halo for the filament. Scale bars 1 μm. Three biological replicates were used for all analyses; Error bars are standard deviations and the center measurement of the graph bars is the mean. The data underlying this figure can be found in S1 Data.
(TIF)

**S3 Fig. Overview pictures of the different labeling strategies. (A)** Field of view (FOV) obtained with a membrane non-permeable dye before the invasion, as illustrated in Fig 2A Top. Image taken after 24 h replication. **(B)** FOV obtained with a membrane-permeable dye before invasion and a second membrane-permeable dye after replication, as illustrated in Fig 2A middle. Image taken after 24 h replication. **(C)** FOV obtained with a membrane nonpermeable dye followed by a membrane-permeable dye before invasion, as illustrated in Fig 2A bottom. Image taken after 24 h replication. All scale bars = 5 μm. **(D)** Comparison of the plasma membrane signal between PM-SAG1 and Int-SAG1 during replication. Int-SAG1 decrease is slower than PM-SAG1. Three biological replicates were used for all analyses; all $P$ values are $0 \le P \le 0.001$, ***, error bars are standard deviations, and the center measurement of the graph bars is the mean. A one-tailed unpaired Student $t$ test was used for all comparisons with no adjustments. The data underlying this figure can be found in S1 Data.
(TIF)

**S4 Fig. Supplementary analysis of triple labeling and strain generation. (A, B)** Quantification of the colocalization between the different populations of SAG1 vesicles as performed in Fig 3C but taking Int-Sag1 (A) and n-SAG1 (B) as reference. **(C)** Integration PCR of the endogenous tagging performed for Figs 3D, 4, and 5. Three biological replicates were used for all analyses; all $P$ values are $0 \le P \le 0.001$, ***, error bars are standard deviations, and the center measurement of the graph bars is the mean. A one-tailed unpaired Student $t$ test was used for all comparisons with no adjustments. The data underlying this figure can be found in S1 Data.
(TIF)

**S5 Fig. Rab5b- mAID-HA and TurboID. (A)** Integration PCR of the mAID tag at the Rab5b locus. **(B)** Full western blot corresponding to the cropped version shown in Fig 4A. **(C)** Integration PCR of the TurboID tag at the Rab5b locus.

**(D)** Representative images of Rab5b-TurboID activity ± biotin. **(E)** Full western blot showing the biotinylation profile of Rab5b-TurboID after various durations of biotin addition. **(F)** Comparison of biotinylation between Rab5b-TurboID and ΔKu80 after 2 h of biotin treatment. **(G)** Streptavidin pull-down comparison between Rab5b-TurboID and ΔKu80. **(H)** Volcano plot generated from mass spectrometry data. **(I)** List of biotinylated proteins enriched specifically in Rab5b-TurboID samples. All experiments were performed in three independent biological replicates. The data underlying this figure can be found in S1 Data.
(TIF)

**S6 Fig. Rab5b- mAID-HA supplementary images. (A)** Representative pictures of the plaque used for the plaque assay analysis in Fig 4B. **(B)** Overexposed version of the pictures used Fig 5G. The PM is still visible in all condition.
(TIF)

**S7 Fig. Evolution of the triple labeling upon MyoF depletion.** Quantification of the colocalization between the different populations of SAG1 vesicles as performed in Fig 4J, but taking Int-Sag1 **(A)** and n-SAG1 **(B)** as reference. White: MyoF ctrl/ − auxin, gray: MyoF KD/ + auxin. Three biological replicates were used for all analyses; all $P$ values are $0 \leq P \leq 0.001$, ***, error bars are standard deviations, and the center measurement of the graph bars is the mean. A one-tailed unpaired Student $t$ test was used for all comparisons with no adjustments. The data underlying this figure can be found in S1 Data.
(TIF)

**S8 Fig. PMR is a dynamic structure. (A, B)** The position of the plasma membrane reservoir (PMR) is not fixed. Fixed imaging of different parasites (A) SAG1-Halo or (B) RH with α-SAG1 in Ultra expansion microscopy. Multiple parasites present a PMR (white arrow) at different positions on the membrane. Scale bar 5 μm for Ultraexpresnion. **(C)** Live microscopy of the PMR dynamics. SAG1-Halo parasites were labeled and transferred to host cells for replication and live imaging. The top of the parasite is highlighted with a yellow arrow. The position of the PMRs is indicated by the white arrows. The position and the size of the PMRs are variable during the replication process. **(D)** Live replication of SAG1-Halo tachyzoite from stage 1 to 4. The PMR is highlighted with white arrows. **(E)** Tachyzoites with daughter cells form without visible PMR. Magenta: SAG1-Halo, Green: IMC1-YFP. **(F)** De novo material is present in the PMR. With the dual labeling with membrane-permeable dye to differentiate between maternal (M-SAG1) and de novo (n-SAG1) SAG1, n-SAG1 was detected to also form the PMR. All scale bars unless stated otherwise = 1 μm.
(TIF)

**S1 Video. SAG1-Halo replicating parasite.** Live microscopy of the plasma membrane reservoir (PMR) formation. SAG1-Halo-expressing parasites were labeled and transferred to host cells for replication and live imaging. The PMR forms at the plasma membrane and is reabsorbed at the end of replication. More than 90% of the parasites imaged formed a PMR. Scale bar = 1 μm.
(AVI)

**S2 Video. The PMR is formed before the IMC of the daughter cells.** Live microscopy of the PMR formation for IMC1-YFP and SAG1-Halo–expressing parasites. The PMR forms prior the apparition of the daughter cells IMC. Scale bar = 1 μm.
(AVI)

**S1 Data. Supporting data.** Excel file containing all numerical values for the figures. The correspondence between the data and the figures is indicated by the name of each sheet.
(XLSX)

**S2 Data. Primer list.** Excel file containing all primers used for strain generation and verification of genetic modifications.
(XLSX)

**S1 Raw Images.** PDF file containing the raw images of the western blots and gels presented in the paper.
(PDF)

## Acknowledgments

We thank Markus Meissner for access to laboratory equipment and microscopes. We are grateful to David Sibley and Aoife Heaslip for providing the Tir1 and MyoF-mAID strains. We acknowledge VEuPathDB for their invaluable informatics resources. Antibodies were kindly provided by David Sibley (Aldolase), Dominique Soldati-Favre (GAP45, MIC4), Lilach Sheiner (G2-Trx), Maryse Lebrun (RON2–4), Vern Carruthers (MIC2), Markus Meissner (MIC8), and Gary Ward (AMA1). We also thank Markus Meissner, Dominique Soldati-Favre, Isabelle Tardieux, Elena Jimenez, Javier Periz, and Mirko Singer for useful discussions, as well as Jennifer Grünert for assistance with electron microscopy.

## Author contributions

**Conceptualization:** Julia von Knoerzer-Suckow, Simon Gras.

**Data curation:** Julia von Knoerzer-Suckow, Eva-Helena Aden, Romuald Haase, Andreas Klingl, Ignasi Forné, Simon Gras.

**Formal analysis:** Julia von Knoerzer-Suckow, Eva-Helena Aden, Romuald Haase, Andreas Klingl, Ignasi Forné, Simon Gras.

**Funding acquisition:** Simon Gras.

**Investigation:** Julia von Knoerzer-Suckow, Eva-Helena Aden, Romuald Haase, Andreas Klingl, Ignasi Forné, Simon Gras.

**Methodology:** Julia von Knoerzer-Suckow, Eva-Helena Aden, Romuald Haase, Andreas Klingl, Ignasi Forné, Simon Gras.

**Project administration:** Simon Gras.

**Supervision:** Simon Gras.

**Validation:** Andreas Klingl, Simon Gras.

**Visualization:** Julia von Knoerzer-Suckow, Romuald Haase, Andreas Klingl, Simon Gras.

**Writing – original draft:** Julia von Knoerzer-Suckow, Simon Gras.

**Writing – review & editing:** Simon Gras.

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
