## [Editor Report · Decision Letter 0]

19 Dec 2024

Dear Simon, 

I hope all is well. Thank you for submitting your manuscript entitled "Toxoplasma gondii micropore is required for regulating the membrane reservoir during parasite replication." for consideration as a Research Article by PLOS Biology.

I could not find an available Academic Editor at the moment, but I have discussed your manuscript within the team and I am writing to let you know that we would like to send your submission out for external peer review.

IMPORTANT: I will be out of office from December 23rd to January 2nd. However, I will do my best to try to recruit reviewers before these dates. Thank you for your patience!

Once your full submission is complete, your paper will undergo a series of checks in preparation for peer review. After your manuscript has passed the checks it will be sent out for review. To provide the metadata for your submission, please Login to Editorial Manager (https://www.editorialmanager.com/pbiology) within two working days, i.e. by Dec 21 2024 11:59PM.

Kind regards,

Melissa

Melissa Vazquez Hernandez, Ph.D.

Associate Editor

PLOS Biology

---

## [Decision Letter · Decision Letter 1]

4 Feb 2025

Dear Dr Gras,

Thank you for your patience while your manuscript "Toxoplasma gondii micropore is required for regulating the plasma membrane reservoir during parasite replication." was peer-reviewed at PLOS Biology. It has now been evaluated by the PLOS Biology editors, an Academic Editor with relevant expertise, and by two independent reviewers. 

In light of the reviews, which you will find at the end of this email, we would like to invite you to revise the work to thoroughly address the reviewers' reports. 

As you will see below, the reviewers are positive about the relevance and novelty of the study, yet some concerns have raised during revision. Reviewer 1 does not understand the conclusions that the study is reaching regarding the fate of plasma membrane reservoir, and think you should clarify the terms PMR vs excess membrane. Reviewer 2 mentions that some findings are confirmatory like the PM is shared between intracellular parasites, and that some conclusions are hard to follow. The reviewer also mentions that the role of Rab5b in the cycle of PM is not really shown, and that the exact function of PMR is not demonstrated, or that a defect in its reabsorption can lead to death in the K13 mutant. After further discussion with the reviewers we think that the study will benefit with additional experiments regarding Rab5b disruption and about PMR, and that the problem regarding PMR is due to lack of functional validation. We agree with all reviewer concerns and would require some additional experimental revisions to address them, as we consider that this would strengthen the work.

Given the extent of revision needed, we cannot make a decision about publication until we have seen the revised manuscript and your response to the reviewers' comments. Your revised manuscript is likely to be sent for further evaluation by all or a subset of the reviewers.

**IMPORTANT - SUBMITTING YOUR REVISION**

*Re-submission Checklist*

*Published Peer Review*

*PLOS Data Policy*

*Blot and Gel Data Policy*

Sincerely,

Melissa

Melissa Vazquez Hernandez, Ph.D.

Associate Editor

PLOS Biology

REVIEWERS' COMMENTS:

Reviewer #1: 

The intracellular protozoan Toxoplasma gondii divides by endodyogeny wherein two daughter cells form within a mother cell. The daughter cells develop their own specialized subpellicular membranes (the inner membrane complex) but utilize the mother cell membrane as the bud from the mother cell. Also, it has been previously established that daughter cells remain connected via a residual body and can share cytosolic contents. These peculiarities open several interesting questions about membrane sources and dynamics during parasite replication. By using thorough and quantitative analyses, the current study discovers that the plasma membrane is shared among parasites within the same replicative compartment (the parasitophorous vacuole), that at each cell division the PM is a hybrid of inherited material, recycled material, and de novo material, and that recycling of the PM is associated with a Rab5b positive compartment. Also, although (as somewhat expected) myoF is necessary for the dynamic movement of endocytic vesicles it is not necessary for endocytosis. The authors also identify a so-called plasma membrane reservoir (PMR) that emanates from the PM and transiently precedes division at each cell cycle, and they suggest this material is reabsorbed in a micropore dependent manner. Overall, this work significantly advancing our understanding of PM dynamics in a system that has unique features compared to model organisms and exposes new biology that is relevant to other protozoa.

Main comments

This reviewer is uncertain of what the authors are suggesting with respect to the fate of PMRs. One the one hand, in the section beginning at line 293, they seem to indicate the PMRs are utilized as a source of membrane to facilitate expansion of the mother cell PM, thereby ensuring sufficient membrane for the budding daughters. This interpretation appears to be supported by video S2. One the other hand, in the section beginning at line 330 they suggest that the PMRs are reabsorbed via the micropore. Although their data shows that impairing micropore function results in more PMR, this is probably because of a block in the normal PM recycling process rather than reabsorption of the PMR itself. In this regard it might be more of an accumulation of excess membrane under a non-natural situation rather than a reabsorption of PMR. Perhaps there needs to be a clarification of terms (PMR vs excess membrane cause be defective recycling) to avoid confusion.

Minor comments

Figure 3d. some of the column labels are cut off.

Reviewer #2: 

Here, the authors examine the dynamic and recycling of the plasma membrane in Toxoplasma gondii. Using FRAP experiments and SAG1-Halo as a proxy for PM dynamics, they nicely showed that the PM of T. gondii has a high fluidity within a parasite, and being also fluid between individual parasites of the same vacuole, which is easily explained by the well-described connection of PM of parasites within a vacuole. They set up a variety of different staining schemes and live-cell imaging to quantify and follow the different pools PM during replication. They observed a very dynamic system, including inheritance of the PM, and both endocytosis and exocytic pathways. Analysis of the potential players of the endocytic pathway, showed partial colocalisation with Rab5B, and a reduction of the traffic of endo-vesicles in absence of MyoF, without any disturbance of PM inheritance or homeostasis. They also described membrane loops, referred as plasma membrane reservoir PMR. This structure is formed transiently, before the start of division, is reabsorbed, translocated to the posterior end of the parasite, where it becomes indistinguishable from the residual body. This structure is proposed to anticipate the future expansion of the PM required during the budding process of the daughter cells. 

Major comments

1. The results and conclusions are sometimes difficult to follow. What is lacking in text and legends is a more detailed description of the experimental steps to help the reader evaluating the authors interpretation of the phenomena. For example, how long after labelling did they analyse parasites? or in figure 3A, what means « begining of intracellular development ? » : is it staining of intracellular parasites or a pulse before invasion (figure 6)? ….

2. Several findings are confirmative. It has been already shown by several EM studies that the PM is shared between intracellular parasites within the same vacuole. What is really new is the analysis of the fluidity of the PM by FRAP experiments. It is also known that the PM follows a cycle of endo- and exocytosis (Gras et al. PloS Biol 2019), and that depletion of K13 reduces the endocytosed vesicles (Koreny et al, Nat Comm 2023).

3. A direct link of the role of Rab5b with the cycle of endo- and exocytosis of PM has not been shown ; they showed only partial colocalization between SAG1 vesicles and Rab5b. The discussion on the potential role of Rab5b is very long, although there is little evidence to support rab5b's role. Are the dynamics of the MP vesicles changed in the rab5b mutant?

4. The exact function of the PMR is not demonstrated, neither that the defect in its reabsorption leads ultimately to the death of the K13 mutant (as mentionned in the abstract). The death of parasites (previously demonstrated in absence of K13) might be due to a defect in endocytosis of key metabolites, as shown recently by Wan et al. Nature Comm 2023. The study is descriptive, the function of PMR remains speculative. If the PMR and its reabsorption serves as a reservoir for PM of daughter cells, why is the PM well visible and looks quite normal in the daugher cells of the K13 mutants ? other potential functions of the PMR might be proposed : a) to supply membrane at the posterior end of parasites for the connection of the daughter cells, which would explain why the K13 mutants are not connected, as shown in Koreny et al. Nat Comm 2023 ; and/or b) a function in establishing the tubulovesicular network that is initiated at the posterior end of the parasite ? 

Other comments

5. For figure 1 (FRAP experiment), the dye used is not indicated in legend and M&M. It is supposed to be a non-permeable one otherwise they would not be able to discreminate between diffusion at PM from neogenesis. 

6. For the measure of neo-SAG1, it is important to control that there is saturation of M-SAG1 with the first dye. In sort, that there is no binding of either external or internal SAG1 pools by subsequent 2nd dye application before replication, such that secondary labelling exclusively labelled the Neo-SAG1 pool. Same apply for the measure of Int-SAG1.

7. Figure 2D, and conclusion lines 186-188. 'we found that loss of intensity after each round of replication is not as severe, when compared to PM-SAG1 (Fig.2E), suggesting that the secreted Int-SAG1 is inserted inside the PM. ' This conclusion is not intuitive and needs to be better explained.

8. The demonstration that parasites share PM continuity within a vacuole is not new. What is really new is the analysis of the fluidity of the PM by FRAP experiments. Change the title of the secetion Line 74 : 'Parasites within a parasitophorous vacuole share a continuous plasma membrane ' to reflect better the novelty of the conclusions of this section.

9. Lines 118-119 : Electron microscopy does not allow to examine if the PM is exchanged between two parasites, it just shows continuity.

10. Figures 1H-J. and corresponding text Lines 121-126 : For non expert in the Toxoplasma field, it is necessary here to introduce what is the residual body, and the presence of F-actin at this location, and what is the Cb-SNAP. Panel H : the enlarged merge does not correspond to the individual enlarged SAG1-HALO, Cb-SNAP or GAP45, or the magnification is not the same for all chanels. The PM covering the residual body is not obvious in the IFA, while it was already well shown by EM, and confirmed here in Fig. 1G. 

11. Line 135 : precise inheritance of the PM in the title 'Inheritance of the maternal material during replication'

12. One can wonder in Fig. 3a, how the authors where able to count the number of vesicles in the residual body, since the labelling appears like a messy staining ?

13. On figure 2C, we see an accumulation of neo-synthetized vesicles in vacuoles containing 8 parasites, while none are visible in the vacuole containing one or two parasites. What does it mean ?

14. Line 194 : mix of Endo-SAG1 AND de novo-SAG1

15. Line 211 : A total of 225 Endo-SAG1, 722 Int-SAG1 and 337 n-SAG1 VESICLES were observed

16. Lines 234-236 : Ref 28 and 29 do not correspond with text

17. in 7b legend line 940 '… with Rab5b appearing to play a role in this process'. The role of Rab5b has not been demonstrated. 

18. in 7a, the successive fluorescence images of a duplication cycle show a PMR in all cases. This is contradictory with the result in fig5 E where the percentage of vacuoles showing a PMR is about 10 %…

---

## [Decision Letter · Decision Letter 2]

8 Sep 2025

Dear Simon,

Thank you for your patience while we considered your revised manuscript "Cycles of growth: plasma membrane recycling and reservoir formation during Toxoplasma gondii replication" for publication as a Research Article at PLOS Biology. This revised version of your manuscript has been evaluated by the PLOS Biology editors, the Academic Editor and one of the original reviewers.

Based on the reviews, we are likely to accept this manuscript for publication, provided you satisfactorily address the remaining points raised by Reviewer one. 

IMPORTANT: Please also make sure to address the following data and other policy-related requests.

1) We routinely suggest changes to titles to ensure maximum accessibility for a broad, non-specialist readership, and to ensure they reflect the contents of the paper. In this case, we would suggest a minor edit to the title, as follows. Please ensure you change both the manuscript file and the online submission system, as they need to match for final acceptance:

"Plasma membrane recycling drives reservoir formation during Toxoplasma gondii intracelullar replication"

2) Please double check the figures. In the last version you uploaded individual figures, and a pdf with the figures. However, several figures did not match and the pdf was missing figures, which made it a bit confusing. For the revision, it would not be necessary to upload the pdf with the figures, and only the individual figures. 

Please supply the numerical values either in the a supplementary file or as a permanent DOI’d deposition for the following figures:

Figure 1CFJ, 2EFG, 3CE, 4CDEF, 5B-FHJ, 6BCDFHJ, 7BE, 8BDEFG, S2C, S3D, S4AB, S5H, S7AB

4) Please cite the location of the data clearly in all relevant main and supplementary Figure legends, e.g. “The data underlying this Figure can be found in S1 Data” or “The data underlying this Figure can be found in https://doi.org/10.5281/zenodo.XXXXX”

5) Please ensure that you are using best practice for statistical reporting and data presentation. These are our guidelines https://journals.plos.org/plosbiology/s/best-practices-in-research-reporting#loc-statistical-reporting and a useful resource on data presentation https://journals.plos.org/plosbiology/article?id=10.1371/journal.pbio.1002128

6) If you are reporting experiments where n ≤ 5, please plot each individual data point.

7) Supplementary files (e.g., excel). Please ensure that all data files are uploaded as 'Supporting Information' and are invariably referred to (in the manuscript, figure legends, and the Description field when uploading your files) using the following format verbatim: S1 Data, S2 Data, etc. Multiple panels of a single or even several figures can be included as multiple sheets in one excel file that is saved using exactly the following convention: S1_Data.xlsx (using an underscore).

8) As some of your findings rely on the microscopy images, we would like to encourage you to submit all other microscopy pictures to Zenodo or FigShare (although this might not be available) so they can be available to the readers too for Figures 1BEGHIK, 2BCD, 3BD, 4BG, 5AGI6AEGI, 7ACDFG, 8AC, S1, S2AB, S3ABC, S5D, S6AB, S8ABCDEF

9) We require the original, uncropped and minimally adjusted images supporting all blot and gel results reported in the Figures 4A, S4C, S5ABCEFG

-- We will require these files before a manuscript can be accepted so please prepare and upload them now. Please carefully read our guidelines for how to prepare and upload this data: https://journals.plos.org/plosbiology/s/figures#loc-blot-and-gel-reporting-requirements

10) Please ensure that your Data Statement in the submission system accurately describes where your data can be found and is in final format, as it will be published as written there

We expect to receive your revised manuscript within two weeks. 

*Published Peer Review History*

*Press*

Sincerely,

Melissa

Melissa Vazquez Hernandez, Ph.D.

Associate Editor

PLOS Biology

REVIEWERS' COMMENTS

Reviewer #1: 

The authors have adequately address the comments of this reviewer. A minor correction is needed i.e., line 235 should be Figure 3B, not Figure 4B.

---

## [Editor Report · Decision Letter 3]

13 Sep 2025

Dear Simon,

Thank you for the submission of your revised Research Article "Plasma membrane recycling drives reservoir formation during Toxoplasma gondii intracellular replication" for publication in PLOS Biology. On behalf of my colleagues and the Academic Editor, Kami Kim, I am pleased to say that we can in principle accept your manuscript for publication, provided you address any remaining formatting and reporting issues. These will be detailed in an email you should receive within 2-3 business days from our colleagues in the journal operations team; no action is required from you until then. Please note that we will not be able to formally accept your manuscript and schedule it for publication until you have completed any requested changes.

PRESS

Sincerely, 

Melissa

Melissa Vazquez Hernandez, Ph.D., Ph.D.

Associate Editor

PLOS Biology
